# HVCN1 Channels Are Relevant for the Maintenance of Sperm Motility During In Vitro Capacitation of Pig Spermatozoa

**DOI:** 10.3390/ijms21093255

**Published:** 2020-05-04

**Authors:** Marc Yeste, Marc Llavanera, Yentel Mateo-Otero, Jaime Catalán, Sergi Bonet, Elisabeth Pinart

**Affiliations:** 1Biotechnology of Animal and Human Reproduction (TechnoSperm), Institute of Food and Agricultural Technology, University of Girona, E-17003 Girona, Spain; marc.yeste@udg.edu (M.Y.); marc.llavanera@udg.edu (M.L.); yentel.mateo@udg.edu (Y.M.-O); sergi.bonet@udg.edu (S.B.); 2Unit of Cell Biology, Department of Biology, Faculty of Sciences, University of Girona, E-17003 Girona, Spain; 3Unit of Animal Reproduction, Department of Animal Medicine and Surgery, Faculty of Veterinary Medicine, Autonomous University of Barcelona, E-08193 Bellaterra (Cerdanyola del Vallès), Spain; dr.jcatalan@udg.edu

**Keywords:** in vitro capacitation, boar sperm, HVCN1 channels, 2-GBI, progesterone

## Abstract

The objective of the present study was to determine the physiological role of voltage-gated hydrogen channels 1 (HVCN1 channels) during in vitro capacitation of pig spermatozoa. Sperm samples from 20 boars were incubated in capacitating medium for 300 minutes (min) in the presence of 2-guanidino benzimidazole (2-GBI), a specific HVCN1-channel blocker, added either at 0 min or after 240 min of incubation. Control samples were incubated in capacitating medium without the inhibitor. In all samples, acrosomal exocytosis was triggered with progesterone after 240 min of incubation. Sperm viability, sperm motility and kinematics, acrosomal exocytosis, membrane lipid disorder, intracellular calcium levels and mitochondrial membrane potential were evaluated after 0, 60, 120, 180, 240, 250, 270 and 300 min of incubation. While HVCN1-blockage resulted in altered sperm viability, sperm motility and kinematics and reduced mitochondrial membrane potential as compared to control samples, at any blocker concentration and incubation time, it had a non-significant effect on intracellular Ca^2+^ levels determined through Fluo3-staining. The effects on acrosomal exocytosis were only significant in blocked samples at 0 min, and were associated with increased membrane lipid disorder and Ca^2+^ levels of the sperm head determined through Rhod5-staining. In conclusion, HVCN1 channels play a crucial role in the modulation of sperm motility and kinematics, and in Ca^2+^ entrance to the sperm head.

## 1. Introduction

Sperm capacitation is characterized by a set of physiological changes preparing the male gamete for fertilization [1]. These physiological changes include increased beat frequency and curvilinear motility, intracellular pH (pH_i_) alkalinization, rise in intracellular calcium levels, augmented membrane lipid disorder and tyrosine phosphorylation of specific proteins, which are required for sperm to trigger acrosomal exocytosis [1,2,3,4,5,6,7,8]. Several channels are implicated in the increase of pH_i_ during capacitation and acrosomal reaction in mammalian spermatozoa, including HCO_3_^−^ membrane transporters, Na^+^-H^+^ exchangers (NHEs), monocarboxylate transporters (MCTs) and voltage-gated proton channels (HVCN1) [9,10]. Nevertheless, the content of these channels in the sperm plasma membrane and their role in pH_i_ regulation differ between mammalian species [9,10,11,12].

HVCN1 channels belong to the superfamily of voltage-gated cation channels [13,14]; they are composed of two subunits, each containing a proton-permeable voltage-sensing domain (VSD), joined by a coiled-coil domain in the C-terminus [15,16,17]. HVCN1 channels differ from other voltage-gated ion channels because they do not have a pore-forming domain; the proton pathway locates within the VSD, thus, each subunit has its own pore and can function independently [15,18,19,20]. HVCN1 channels drive protons more quickly and efficiently than transporters or exchangers do, and lead them unidirectionally to the extracellular medium [12].

Interestingly, not only does HVCN1 gating rely on membrane voltage, but also on the pH difference (ΔpH) across the plasma membrane [21]. Therefore, when changes in intracellular (pH_i_) and extracellular pH (pH_o_) are not associated with changes in ΔpH (pH_o_–pH_i_ = 0), the voltage dependence of HVCN1-activation is low; conversely, when changes result in ΔpH > 0 or ΔpH < 0, its voltage activation is high [22]. Therefore, the voltage activation threshold of HVCN1 channels is dependent on the ΔpH across the plasma membrane [23].

HVCN1 channels have an essential regulating role in several cell types (reviewed in [24]). Nevertheless, little data exist on their physiological role in sperm cells, and most of these studies have been performed in humans. In human sperm, HVCN1 channels regulate sperm motility and are involved in oocyte fertilization [21,23]. Mature sperm stored in the epididymal cauda remain quiescent because pH_i_ and pH_o_ are acidic and ΔpH = 0; upon ejaculation, quiescent sperm mixes with alkaline seminal plasma, which leads to gating of HVCN1-channels and activates sperm motility. In addition, differences between pH_i_ and pH_o_ also occur during the sperm transit throughout the female reproductive tract. At the site of fertilization, HVCN1 channels show enhanced activity, which leads to the increased pH_i_ observed in capacitated spermatozoa (reviewed in [21,23]).

While HVCN1 channels have been identified in pig sperm [21], their physiologic relevance is unknown. Therefore, the present study aimed at investigating the physiological role of those channels during in vitro capacitation of pig sperm. This functional approach was carried out pharmacologically through using 2-guanidino benzimidazole (2-GBI), a specific blocker of HVCN1 channels that inhibits their proton conductance via binding the intracellular VSD domain in the opened channel [12,17].

## 2. Results

Sperm samples were incubated in in vitro capacitating medium for 300 min. After 240 min of incubation, progesterone was added to induce sperm hyperactivation and acrosomal exocytosis. To address the physiological role of HVCN1 channels during in vitro capacitation of pig sperm, some samples were incubated in the presence of 2-GBI blocker at 1, 5 or 10 mM added at time 0 (Experiment 1). Moreover, and in order to understand the functional relationship between progesterone and HVCN1 channels, 2-GBI blocker was added together with progesterone at 240 min of incubation, in a second group of samples (Experiment 2).

### 2.1. Identification of HVCN1 Channels

In order to confirm the presence of HVCN1 channels in pig sperm, an immunoblotting assay was performed. The results showed the presence of an HVCN1-specific band of 70 kDa in ejaculated sperm samples and an unspecific band of 45 kDa. A peptide competition assay proved that the 70 kDa-band appearing in sperm samples was specific for HVCN1 (Figure 1).

### 2.2. Sperm Motility

#### 2.2.1. Total and Progressive Sperm Motility

In experiment 1, the percentages of total and progressively motile sperm were significantly lower (*p* < 0.05) in samples incubated in the presence of 2-GBI than in control samples throughout the 300 min of incubation (Figure 2A–B). The addition of progesterone to control samples at 240 min led to a significant increase (*p* < 0.05) in the percentages of total and progressively motile sperm after 270 min of incubation. Interestingly, the addition of progesterone to samples incubated with 2-GBI had no effect on total and progressive sperm motility.

In experiment 2, the joint addition of progesterone and 2-GBI (acute blocked samples) after 240 min of incubation resulted in a significant decrease (*p* < 0.05) in the percentages of total and progressively motile sperm (Figure 2C–D). The decrease in these two sperm motility parameters was dependent on 2-GBI concentration (*p* < 0.05).

#### 2.2.2. Kinematic Parameters of Curvilinear Velocity (VCL), Straight-Line Velocity (VSL) and Average Path Velocity (VAP)

In experiment 1, sperm kinematic parameters were significantly higher (*p* < 0.05) in control samples than in those incubated with 2-GBI throughout the experimental period. The addition of progesterone to control samples resulted in a slight but significant (*p* < 0.05) increase in VCL, VSL and VAP (Figure 3A–C). In contrast, the addition of progesterone to samples incubated with 2-GBI led to a non-significant increase in VCL, VSL and VAP at 270 min.

In experiment 2, the joint addition of progesterone and 2-GBI (acute blocked samples) after 240 min of incubation resulted in a significant (*p* < 0.05) decrease in sperm kinematic parameters, irrespective of inhibitor concentration (Figure 3D–F).

#### 2.2.3. Indexes of Linearity (LIN), Straightness (STR) and Oscillation (WOB)

In experiment 1, indexes of LIN, STR and WOB were significantly lower in samples incubated with 2-GBI than in the control (*p* < 0.05). The addition of progesterone after 240 min of incubation had a non-significant effect on those indexes in both control and blocked samples (Figure 4A–C).

In experiment 2, the joint addition of progesterone and 2-GBI resulted in a significant decrease (*p* < 0.05) of LIN, STR and WOB indexes in a blocker dose-dependent manner (Figure 4D–F).

#### 2.2.4. Amplitude of Lateral Head Displacement (ALH) and Beat Cross Frequency (BCF)

In experiment 1, ALH and BCF were significantly lower in samples incubated with 2-GBI than in the control (*p* < 0.05). The addition of progesterone had a non-significant effect on these kinematic parameters in control samples and in those incubated with 1 mM and 5 mM 2-GBI. However, incubating samples with 1 and 5 mM 2-GBI resulted in a significant (*p* < 0.05) increase of ALH at 300 min (Figure 5A–B).

In experiment 2, the joint addition of progesterone and 2-GBI (acute blocked samples) led to a significant (*p* < 0.05) decrease in ALH after 250 min of incubation, at any blocker concentration. BCF also dropped (*p* < 0.05) after 250 min of incubation but in a dose-dependent manner (Figure 5C–D).

### 2.3. Sperm Viability

In control samples, sperm viability decreased significantly (*p* < 0.05) at 120 min and 240 min, reaching a value that maintained without significant differences until the end of the incubation. In experiment 1, the presence of 2-GBI blocker resulted in a significant decrease (*p* < 0.05) in sperm viability over the first 180 min of incubation in a dose-dependent manner (Figure 6A); after this decrease, changes in sperm viability were not significant at any blocker concentration.

In experiment 2, the joint addition of progesterone and 1 mM 2-GBI did not affect sperm viability (Figure 6B), whereas the joint addition of progesterone and 5 mM and 10 mM 2-GBI resulted in a significant decrease (*p* < 0.05) of this parameter after 270 min and 250 min of incubation, respectively.

### 2.4. Acrosomal Exocytosis

In experiment 1, the presence of 2-GBI blocker resulted in a significant increase (*p* < 0.05) in the percentage of viable sperm with an exocytosed acrosome from the first 60 min of incubation and in a dose-dependent manner (Figure 7A). The addition of progesterone induced acrosomal exocytosis in viable sperm (*p* < 0.05) in both control and blocked samples.

In experiment 2, the joint addition of progesterone and 2-GBI led to a significant (*p* < 0.05) increase in the percentages of viable sperm with an exocytosed acrosome from 250 min of incubation. The higher concentration of the blocker, the higher the extent of the increase (*p* < 0.05) in the percentage of viable sperm with an exocytosed acrosome (Figure 7B).

### 2.5. Membrane Lipid Disorder

In all samples, membrane lipid disorder of viable sperm increased progressively during the first 120 min of incubation (Figure 8). In experiment 1, this increase was significantly (*p* < 0.05) higher in blocked than in control samples. In control samples, the addition of progesterone resulted in a significant (*p* < 0.05) increase in the percentage of viable sperm with high membrane lipid disorder from 250 min of incubation. In contrast, the addition of progesterone had a non-significant effect in blocked samples, regardless of 2-GBI concentration (Figure 8A).

In experiment 2, the addition of progesterone together with that of 1 mM 2-GBI induced a significant increase (*p* < 0.05) in the percentage of viable sperm with high membrane lipid disorder at 270 min (Figure 8B). The joint addition of progesterone and 5 mM or 10 mM 2-GBI resulted in a rise in the percentage of viable sperm with high lipid membrane disorder at 250 min (*p* < 0.05).

### 2.6. Intracellular Calcium Levels

#### 2.6.1. Fluo3 Staining

In experiment 1, around 50% of viable sperm showed positive Fluo3-staining (Fluo3^+^) during the first 240 min of incubation in control and blocked samples (Figure 9). The addition of progesterone resulted in a significant increase (*p* < 0.05) in the percentages of viable sperm showing Fluo3^+^ staining in the control and samples incubated with 1 mM 2-GBI at 250 min. Samples incubated with 5 mM and 10 mM 2-GBI showed a delayed but significant increase (*p* < 0.05) in the percentages of viable sperm exhibiting Fluo3^+^ staining at 300 min (Figure 9A). On the other hand, the fluorescence intensity of Fluo3^+^ in viable sperm maintained without significant variations over the first 240 min of incubation in all samples (Figure 9B). The addition of progesterone resulted in a significant increase (*p* < 0.05) in the fluorescence intensity of Fluo3^+^ (viable sperm) in control and samples incubated with 1 mM and 5 mM 2-GBI, but it did not affect that intensity in those incubated with 10 mM 2-GBI.

In experiment 2, the addition of progesterone to control samples and the joint addition of progesterone and 1 mM 2-GBI led to a significant (*p* < 0.05) increase in the percentages of viable sperm showing Fluo3^+^ staining at 250 min (Figure 9C). In contrast, the joint addition of progesterone and 5 mM or 10 mM 2-GBI resulted in a delayed but significant (*p* < 0.05) increase in those percentages at 270 min. Moreover, fluorescence intensity of Fluo3^+^ in viable sperm was significantly lower (*p* < 0.05) in blocked than in control samples after 250 min of incubation (Figure 9D). At 270 min, it was inversely dependent on blocker concentration; therefore, the higher the concentration of 2-GBI, the lower the fluorescence intensity. Finally, at 300 min of incubation, fluorescence intensity did not differ between the control and samples incubated with 1 mM or 5 mM 2-GBI, but it was significantly lower (*p* < 0.05) in samples blocked with 10 mM 2-GBI.

#### 2.6.2. Rhod5 Staining

In experiment 1, percentages of viable sperm with Rhod5^+^ staining were significantly higher (*p* < 0.05) in blocked than in control samples throughout the entire experimental period (Figure 10A). After 60 min of incubation, the higher the blocker concentration, the higher the percentage of viable sperm showing Rhod5^+^ staining (*p* < 0.05); however, no significant differences between blocking treatments were observed from 120 min until the end of the experiment. Whereas the addition of progesterone had little effect on sperm incubated with 2-GBI, it led to a significant increase (*p* < 0.05) in the percentage of viable sperm exhibiting Rhod5^+^ staining in control samples. The fluorescence intensity of Rhod5^+^ (viable sperm) showed no significant differences between control and blocked samples in nearly all time points, despite being significantly higher at both 180 and 300 min of incubation (*p* < 0.05) in blocked samples with 10 mM 2-GBI (Figure 10B).

In experiment 2, percentages of viable sperm with Rhod5^+^-staining raised significantly (*p* < 0.05) after 250 min of incubation in both control and acute blocked samples, the differences between treatments being not significant (Figure 10C). At 270 min, this percentage dropped significantly (*p* < 0.05) in all samples; the extent of that decrease, however, was higher in samples incubated with 5 mM and 10 mM 2-GBI than in the control and samples incubated with 1 mM 2-GBI. At the end of the experiment, the percentage of viable sperm with Rhod5^+^-staining was significantly higher (*p* < 0.05) in the control than in samples blocked with 5 mM and 10 mM 2-GBI. The fluorescence intensity of Rhod5^+^ in viable sperm was significantly higher (*p* < 0.05) in samples incubated with 1 mM and 5 mM 2-GBI and significantly lower (*p* < 0.05) in samples incubated with 10 mM 2-GBI after 250 min of incubation, but it did not differ between treatments at 270 min (Figure 10D). At 300 min, the fluorescence intensity was significantly lower (*p* < 0.05) in samples incubated with 1 mM 2-GBI than in the control and samples blocked with 5 mM and 10 mM 2-GBI.

### 2.7. Mitochondrial Membrane Potential

In experiment 1, the percentages of sperm with high mitochondrial membrane potential were significantly higher (*p* < 0.05) in the control and in samples incubated with 1 mM 2-GBI than in those incubated with 5 mM and 10 mM 2-GBI throughout the entire experimental period (Figure 11A). In control samples, the addition of progesterone after 240 min of incubation resulted in a peak (*p* < 0.05) in this percentage at 250 min and 270 min. In samples incubated with 2-GBI, a delayed increase (*p* < 0.05) in the percentages of sperm with high mitochondrial membrane potential was observed at 270 min. The extent of that increase depended on the inhibitor concentration; thus, the higher the 2-GBI concentration, the lower the increase in the mitochondrial membrane potential (*p* < 0.05). An effect of 2-GBI concentration was also observed in the orange (FL2) fluorescence intensity of sperm with high membrane mitochondrial potential, with significantly higher (*p* < 0.05) values of this parameter in the control samples than in samples incubated with 10 mM 2-GBI (Figure 11B). While the addition of progesterone after 240 min of incubation resulted in a significant increase (*p* < 0.05) in the orange (FL2) fluorescence intensity of sperm with high membrane mitochondrial potential in both the control and samples incubated with 1 mM 2-GBI, it had little effect on those incubated with 5 mM or 10 mM 2-GBI.

In experiment 2, the addition of progesterone and the joint addition of progesterone and 2-GBI resulted in a significant increase (*p* < 0.05) in the percentages of sperm with high mitochondrial membrane potential at 250 min in both control and acute blocked samples (Figure 11C). Nevertheless, at inhibitor concentrations of 5 mM and 10 mM, this increase was followed by a significant decrease (*p* < 0.05) after 270 min of incubation. The increase in the mitochondrial membrane potential was also apparent from the augmented orange (FL2; JC1_agg_) fluorescence intensity of sperm with high membrane mitochondrial potential observed in both the control and samples incubated with 1 mM of 2-GBI (*p* < 0.05) (Figure 11D). Conversely, the orange (FL2) fluorescence intensity of sperm with high membrane mitochondrial potential decreased significantly (*p* < 0.05) when samples were added together with progesterone and 5 mM and 10 mM 2-GBI.

## 3. Discussion

Recent evidence supports the relevance of ion channels in orchestrating the sequence of events associated with sperm capacitation [5,7,25]. However, differences in the content and regulation mechanisms of these ion channels exist between mammalian species, thus suggesting a species-specific mechanism of sperm capacitation regulation [5,7,25,26,27]. While HVCN1 channels are not present in mouse sperm [21,23], they have been reported to be present in human and bull sperm, showing a molecular weight of 70–73 kDa (32–35 kDa per monomer). Prior to our work, there was only one study demonstrating the presence of HVCN1 channels in the plasma membrane of boar sperm [21]. The 35 kDa-band reported by Berger et al. [21] corresponds with the molecular mass of HVCN1 monomers. These results are consistent with the 70 kDa-band observed in our immunoblotting assays, since HVCN1 channels are known to be found in cells as dimers [28]. In addition, it is worth highlighting that the 70 kDa-band observed in our immunoblotting assays disappeared when peptide blocking experiments were performed, thus indicating that the band specifically corresponded to the HVCN1 channel.

It is worth mentioning that the changes observed in plasma membrane integrity and permeability, sperm motility and kinematics, mitochondrial membrane potential and intracellular calcium levels observed herein during in vitro capacitation and after progesterone-induced acrosomal exocytosis are in line with previously reported studies [2,6,7,29,30,31,32]. In addition, the present study has demonstrated, for the first time, the physiological relevance of HVCN1 channels during in vitro capacitation and progesterone-induced acrosomal exocytosis of pig sperm. We found that inhibiting HVCN1 channels at 0 min and after 240 min of incubation reduced the sperm viability and impaired the total and progressive motility of in vitro capacitated pig sperm. Blocking HVCN1 channels has also been reported to result in decreased sperm viability in human [33] and bull spermatozoa [10], due to their inability to regulate their pHi [23]. Since only viable spermatozoa are able to fertilize the oocyte, in the present work, we have analyzed the physiological changes occurring in the viable sperm population with blocked HVCN1 channels in order to determine the real change of viable spermatozoa to undergo the sequence of events associated with in vitro capacitation and subsequent fertilization ability.

In our study, the sperm motility decrease was concomitant with a reduction in sperm velocity and linearity, regardless of blocker concentration and time of addition. Similar results were obtained in human sperm after HVCN1 inhibition with Zn [33]. In bull [10] and human sperm [11], HVCN1 channels are considered to be essential for the activation of progressive motility after ejaculation and in the regulation of hypermotility during capacitation. In both non-capacitated and capacitated human sperm, the intracellular alkalinization is critical for sperm motility activation, since it triggers the sperm-specific soluble adenylyl cyclase/protein kinase A pathway (sAC/cAMP/PKA), which stimulates cell metabolism and propels the axoneme (reviewed in [11]). In humans, HVCN1 activity is higher in capacitated than in non-capacitated sperm due the phosphorylation of the channel during capacitation [11,23]. In this work, the decrease in sperm motility and kinematics observed in samples incubated with the inhibitor (2-GBI) correlated with the reduction in mitochondrial membrane potential. These findings suggest that HVCN1 channels are essential for pig sperm to increase their oxidative phosphorylation during sperm capacitation and acrosomal exocytosis induced by progesterone. In agreement with our results, Musset et al. [34] reported that HVCN1 channels are involved in the generation of superoxide radicals in human sperm, which are reactive oxygen species resulting from the activity of the mitochondrial membrane chain. In humans, alterations in mitochondrial membrane potential observed after HVCN1 inhibition are also associated with the decrease of sperm viability [33].

The inhibitory effect of 2-GBI was higher when added at 0 min than when added after 240 min of incubation. In addition, while the blocking effects of 2-GBI did not rely on its concentration when added at 0 min, they were dose-dependent when added at 240 min. These findings agree with the blocking mechanism of this inhibitor, which binds the intracellular VSD domain of HVCN1 when channels are opened, thereby preventing H^+^ efflux [12,17]. When added at 0 min, the number of blocked HVCN1 channels increases progressively throughout the incubation period, all being nearly blocked at 240 min. This could explain the absence of a dose-dependent effect on nearly all sperm motility and kinematic parameters, and the inability of progesterone to hyperactivate sperm motility. In contrast, the reduced and dose-dependent effect observed when 2-GBI was added at 240 min indicates that fewer HVCN1 channels were blocked. In these acute blocked samples, not only was progesterone able to increase sperm motility, but the extent of that increase also depended on the inhibitor concentration.

During capacitation, HVCN1 channels have been suggested to induce hypermotility through the cAMP/PKA pathway in bull sperm [10], and via activating CatSper channels in human sperm [11,23,33,35]. In the present study, we found that HVCN1 activation during in vitro capacitation of pig sperm regulates Ca^2+^ entrance to the sperm tail, determined by Fluo3 staining, despite the physiological relationship between both processes still being unclear. It is worth mentioning that although blocking HVCN1 channels at the beginning of the experiment (i.e., 0 min) did not affect the Ca^2+^ influx determined through Fluo3 staining (sperm tail) during the first 240 min incubation, samples blocked with 5 and 10 mM 2-GBI showed a defective Ca^2+^ entrance after progesterone addition. This effect was not observed when the inhibitor was added at 1 mM (Experiment 1). Moreover, when added together with progesterone at 240 min of incubation, 2-GBI blocked Ca^2+^ influx at 10 mM, but not at 1 mM or 5 mM (Experiment 2). Moreover, in samples blocked at 0 min and at 240 min (acute), the pattern of variations in Ca^2+^ levels determined through Fluo3 did not agree with that of sperm motility and velocity. Altogether, our results suggest that while HVCN1 channels are essential for pig sperm motility during in vitro capacitation, they do not appear to be essential in the regulation of Ca^2+^ influx. Despite further research being required, we hypothesize that Ca^2+^ entrance to the flagellum in capacitated pig sperm does not rely upon the activity of HVCN1 channels, but rather depend on a direct and non-direct effect of progesterone on Ca^2+^ channels.

In rodents [36], cattle [37] and humans [11], intracellular alkalinization has been correlated with the activation CatSper channels and the rise in intracellular Ca^2+^ levels during sperm capacitation. Moreover, in human sperm, HVCN1 and CatSper channels are localized in the same plasma membrane domains, thus evidencing their strong functional relationship. In our study, however, the absence of a strong relationship between HVCN1 activity and Ca^2+^ influx determined through Fluo3 suggests that not only is intracellular alkalinization regulated by HVCN1 channels in boar sperm, but also by other H^+^ transporters. These differences reflect the complex and species-specific regulation of sperm physiology.

Little data exist about the effects of HVCN1 blockage on acrosomal exocytosis in pigs and other mammalian species. The present study showed that HVCN1 inhibition had a different effect on acrosomal exocytosis depending on whether the 2-GBI blocker was added at the beginning of the experiment (i.e., 0 min) or after 240 min of incubation. In sperm incubated with 2-GBI from time 0 (experiment 1), acrosomal exocytosis was higher than in control samples, and was associated with high plasma membrane lipid disorder (measured by M540 staining) and increased Ca^2+^ levels evaluated through Rhod5 staining, which has an affinity for the Ca^2+^ residing in the sperm head. While in these samples, in which HVNC1 channels were inhibited by 2-GBI, the addition of progesterone acted as a potent acrosomal exocytosis inducer, neither a new rise in the intracellular Ca^2+^ levels stained by Rhod5 nor a further increase in membrane lipid disorder were observed. In addition, the premature acrosomal exocytosis observed in this experiment was concomitant with a premature rise in Ca^2+^ levels and cholesterol efflux. Related to this, a close relationship between pH_i_ and the cholesterol content of the plasma membrane has been reported in human sperm [38]. Moreover, recent studies in rodents have demonstrated that the sterol efflux induces a local depolarizing effect on the plasma membrane that may activate different types of transient voltage-gated cation channels, leading to Ca^2+^ rises and triggering acrosomal exocytosis [39,40]. In contrast, when added at 240 min together with progesterone (experiment 2), 2-GBI blocker had little effect on acrosomal exocytosis and plasma membrane lipid disorder, and led to reduced Ca^2+^ levels determined by Rhod5 (sperm head) as compared to control samples. Moreover, zinc blocking of HVCN1 channels in capacitated human sperm reduces progesterone-induced acrosomal exocytosis due to the inhibition of CatSper channels, which again underpins the close functional relationship between these two kinds of channels in this species [33]. In capacitated pig spermatozoa, SLO1 blockage also inhibits progesterone-induced acrosomal exocytosis by reducing Ca^2+^ entrance to the sperm head [7], whereas the presence of zinc prevents acrosomal membrane modifications associated with acrosomal exocytosis [41].

As previously discussed for sperm motility and kinematic parameters, the differences observed between Experiment 1 and Experiment 2 in acrosomal exocytosis are likely to be related to the number of blocked HVCN1 channels. The alterations observed after total blockage in Experiment 1 suggest that HVCN1 channels are essential to maintain pig sperm homeostasis during in vitro capacitation and for preventing premature sperm activation. From Experiment 2, one can conclude that while HVCN1 channels are relevant for the rise in Ca^2+^ levels (Rhod5 staining) observed after progesterone addition, they are not specifically involved in the mechanism that triggers acrosomal exocytosis. On the other hand, another interesting finding of our study was that HVCN1 blockage affected differently the intracellular Ca^2+^ levels determined by Fluo3 (which has more affinity for the Ca^2+^ residing in the tail) and those assessed by Rhod5 (which has more affinity for the Ca^2+^ head store), thereby indicating that the involvement of HVCN1 channels in regulating these two stores is distinct. These results are in line with previous studies performed in human [42], mouse [43] and pig sperm [7].

Despite alkalinization being considered an essential process in sperm capacitation, little data are available about changes in pH_i_ and their effects on sperm physiology, as well as on the types of channels and transporters implicated in this pH_i_ increase in different mammalian species [44]. In human sperm, HVCN1 channels modulate sperm motility and are involved in membrane potential changes during hyperpolarization [11,45]. In the present study, we have demonstrated the relevance of HVCN1 channels for sperm motility and kinematics during in vitro capacitation of pig sperm. Nevertheless, further research is necessary to elucidate the physiological role of these channels in pig sperm.

## 4. Materials and Methods

### 4.1. Materials

All chemicals were purchased from Sigma-Aldrich Química (Madrid, Spain) unless otherwise indicated.

### 4.2. Semen Samples

The study was performed using commercial seminal doses from 20 Piétrain boars, 10 boars per experiment, ranging from 18 to 24 months of age. Boars were kept under standard conditions of temperature and humidity, fed a standard diet and provided with water *ad libitum*, following the guidelines for animal welfare and handling established by the Animal Welfare Regulations issued from the Regional Government of Catalonia (Spain). Since semen samples were provided by a local farm (Selecció Batallé S.A.; Riudarenes, Spain) that operates under commercial, standard conditions and authors did not manipulate any animal, no specific approval from an Ethics Committee was required.

Ejaculates were collected twice a week using the gloved-hand technique, and the sperm-rich fraction was immediately filtered through gauze to remove the gel, and pre-diluted 2:1 (v:v) in a long-term extender (Vitasem, Magapor, Ejea de los Caballeros, Zaragoza, Spain) at 37 °C inside a collecting recipient. Commercial seminal doses were obtained after packaging diluted sperm-rich fractions into 90 mL bags at a concentration of 3 × 10^9^ sperm/dose. Commercial seminal doses were then cooled down to 16 °C, and three doses per collection and boar were sent to our laboratory in a heat-insulating container at 16 °C. All seminal doses used in the present study had a sperm quality above the following thresholds: 70% of total motile sperm, 80% of viable sperm and 85% of morphologically normal sperm (data not shown).

### 4.3. Experimental Design

In each boar, the presence of HVCN1 channels in sperm plasma membrane was checked by immunoblotting. The physiological role of HVCN1 channels during in vitro capacitation was determined by incubating sperm samples in capacitating medium and in the presence of 2-guanidino benzimidazole (2-GBI), a specific inhibitor of HVCN1 channels. This blocking agent was added either at the beginning of the experiment (0 min; experiment 1) or after 240 min of incubation (experiment 2), and three separate concentrations (1 mM, 5 mM and 10 mM), set following preliminary experiments and after reviewing the literature [16], were tested.

In the first experiment, three seminal doses of each boar were pooled, distributed into 28 aliquots of 8 mL each, and centrifuged at 600 × g and 16 °C for 5 min. Sperm pellets were immediately resuspended in capacitating medium (20 mM HEPES, 112 mM NaCl, 3.1 mM KCl, 5 mM glucose, 21.7 mM sodium-L-lactate, 1 mM sodium pyruvate, 0.3 mM Na_2_HPO_4_, 0.4 mM MgSO_4_·7H_2_O, 4.5 mM CaCl_2_·2H_2_O, 5 mg/mL bovine serum albumin (BSA) and 15 mM sodium bicarbonate) to a final concentration of 1 × 10^7^ sperm/mL. Aliquots were distributed into control samples (seven aliquots) and blocked samples (21 aliquots), the latter being incubated in the presence of 2-GBI at different concentrations (1 mM, 5 mM or 10 mM). In all blocked samples, 2-GBI was added at 0 min, just before starting the experiment. Samples were incubated together at 38.5 °C, 100% humidity and 5% CO_2_ in a Hera Cell 150 incubator (Heraeus, Germany) for 60 min, 120 min, 180 min, 240 min, 250 min, 270 min or 300 min. In samples incubated for 250 min, 270 min and 300 min, 10 µg/mL progesterone was added at 240 min.

In the second experiment, seminal doses were divided into 16 aliquots of 8 mL each, centrifuged at 600 × g and 16 °C for 5 min and resuspended in capacitating medium to a final concentration of 1 × 10^7^ sperm/mL. Four aliquots were incubated at 38.5 °C and 5% CO_2_ for 60 min, 120 min, 180 min or 240 min, whereas the other 12 aliquots were incubated for 250 min, 270 min or 300 min. These 12 aliquots were divided into control and acute blocked samples. After 240 min of incubation, control samples were added with 10 µg/mL progesterone, whereas the others (acute blocked samples) were added with progesterone (10 µg/mL) together with 1 mM, 5 mM or 10 mM 2-GBI.

Physiological changes of sperm samples were determined at each relevant time point, i.e., 0 min, 60 min, 120 min, 180 min, 240 min, 250 min, 270 min or 300 min, by evaluating sperm motility and kinematics, sperm viability, membrane lipid disorder, acrosomal exocytosis, intracellular calcium levels and mitochondrial membrane potential.

### 4.4. Immunoblotting

Semen samples were centrifuged and resuspended in cell lysis buffer (Tractor^TM^ Buffer, Takara). Next, samples were incubated in agitation at 4 °C for 30 min and centrifuged at 12,000 × g and 4 °C for 15 min. Finally, supernatants were quantified for total protein through a detergent compatible method (DC™ Protein Assay, Bio-Rad, Hercules, CA, USA).

Twenty µg of total protein were resuspended in reducing buffer (Sample Buffer, Merk, Darmstadt, Germany) and incubated at 95 °C for 5 min prior to loading onto a 12% polyacrylamide gel (Mini-PROTEAN^®^ TGX Stain-Free™ Precast Gels, Bio-Rad). Following this, total protein bands were visualized by UV exposition and acquisition using a G:BOX Chemi XL system (SynGene, Frederick, MD, USA). This method is based on the fluorescent detection of tryptophan residues modified by a trihalo compound present in the gel. Then, proteins from gels were transferred onto polyvinylidene fluoride (PVDF) membranes using Trans-Blot^®^ Turbo^TM^ (Bio-Rad).

Subsequently, membranes were blocked for 1 h at room temperature (RT) with agitation in blocking solution (10 mmol/L Tris, 150 mmol/L NaCl, and 0.05% Tween-20; pH = 7.3, and 5% bovine serum albumin, Roche Diagnostics, S.L., Basel, Switzerland). Blocked membranes were incubated overnight at 4 °C with an anti-HVCN1 rabbit polyclonal antibody (ref. LS-C146080, LifeSpan BioSciences, Seattle, WA, USA; 1:10,000, v:v), rinsed thrice and incubated with an anti-rabbit horseradish peroxidase (HRP)-conjugated antibody (ref. P0448, Agilent, Santa Clara, CA, USA; 1:20,000, v:v) for 1 h at RT. Subsequently, membranes were washed thrice, and bands were visualized with a chemiluminescent substrate (Immobilon ECL Ultra Western HRP Substrate, Merk). Finally, membranes were scanned using G:BOX Chemi XL. Next, membranes were stripped by incubation at RT in agitation for 10 min with a stripping buffer (0.2 mol/L glycine (Serva), 0.1% (w:v) SDS and 1% (v:v) Tween20; pH adjusted to 2.2) and blocked in blocking solution at RT for 1 h. Following this, membranes were incubated with the loading control (anti-α-tubulin monoclonal mouse antibody; ref. MABT205, Millipore, Temecula, CA, USA; 1:100,000, v:v) at RT for 1 h. Thereafter, membranes were washed, incubated with a secondary anti-mouse HRP-conjugated antibody (ref. P0260, Agilent; 1:200,000, v:v) at RT for 1 h and incubated with the HRP substrate prior to their scanning with the G:BOX Chemi XL system.

The specificity of the anti-HVCN1 antibody was confirmed through peptide competition assays utilizing HVCN1 blocking peptide (ref. LS-E12579, LifeSpan BioSciences), 50 times in excess regarding the antibody (Figure 1).

### 4.5. Evaluation of Sperm Motility and Kinematics

To measure sperm motility and kinematics, a 5-µL droplet was mounted onto a pre-warmed (38 °C) Makler chamber (Sefi-medical Instruments, Haifa, Israel). Sperm motility was examined at 100 × magnification using a negative phase-contrast objective coupled to an Olympus BX41 microscope (Olympus Europe GmbH, Hamburg, Germany), and equipped with ISAS software (ISAS Ver. 1.0; Proiser S.L., Valencia, Spain). In all samples (control, 1 mM 2-GBI, 5 mM 2-GBI and 10 mM 2-GBI blocked samples, and 1 mM 2-GBI, 5 mM 2-GBI and 10 mM 2-GBI acute blocked samples), three replicates of at least 1000 sperm per replicate were examined. In each field, 25 consecutive digitalized frames per second were acquired to assess sperm motility and kinematics. A spermatozoon was considered to be motile when its VAP (velocity of the average pathway) was equal to or higher than 10 µm/s, and progressively motile when its percentage of straightness (STR) was equal to or higher than 45% [6]. For each incubation time and treatment, sperm motility was expressed as percentages of total and progressively motile spermatozoa (data are shown as mean ± SEM; *n* = 10; Figure 2).

Sperm kinematics was analyzed from the evaluation of velocity parameters, i.e., curvilinear velocity (VCL, average velocity measured over the actual point-to-point track followed by the sperm head; µm/s); straight-line velocity (VSL, average path velocity of the sperm head along a straight line from its first to its last position; µm/s); average path velocity (VAP, velocity of the sperm head along its average trajectory; µm/s); and the evaluation of indexes of linearity (LIN: ratio between VSL and VCL; %), straightness (STR: ratio between VSL and VAP; %) and oscillation (WOB: ratio between VAP and VCL; %). The amplitude of lateral head displacement (ALH: average value of the extreme side-to-side movement of the sperm head in each beat cycle; µm) and beat cross frequency (BCF: frequency with which the actual sperm trajectory crosses the average path trajectory; Hz) were also recorded. Results are expressed as mean ± SEM (*n* = 10; Figure 3, Figure 4 and Figure 5).

### 4.6. Flow Cytometry Analyses

Sperm viability, lipid disorder of plasma membrane, acrosomal exocytosis, intracellular calcium levels and mitochondrial membrane potential were determined through flow cytometry. In each assay, sperm concentration was first adjusted to 1 × 10^6^ sperm/mL in Beltsville Thawing Solution (BTS) in a final volume of 0.5 mL. Three replicates per incubation time, experiment, 2-GBI concentration and sperm parameter were examined in a Cell Laboratory QuantaSC™ cytometer (Beckman Coulter; Fullerton, CA, USA).

Information about flow cytometry analyses conducted in this work is given according to the recommendations of the International Society for Advancement of Cytometry (ISAC). Samples were excited with an argon ion laser (488 nm) set at a power of 22 mW. For each particle, characteristics were evaluated and plotted as Electronic Volume (EV, equivalent to Forward Scatter) and Side Scatter (SSC). Three optical filters were used with the following optical properties: FL1 (green fluorescence): Dichroic/Splitter, DRLP: 550 nm, BP filter: 525 nm, detection width 505–545 nm; FL2 (orange fluorescence): DRLP: 600 nm, BP filter: 575 nm, detection width: 560–590 nm; and FL3 (red fluorescence): LP filter: 670, detection width: 670 ± 30 nm). Flow rate was set at 4.17 μL/min in all analyses and a minimum of 10,000 events per replicate was evaluated. The analyzer threshold was adjusted on the EV channel to exclude subcellular debris (particles diameter <7 μm) and cell aggregates (particles diameter >12 μm). The sperm-specific events were positively gated on the basis of EV/SSC distributions. The other events were gated out.

In all flow cytometry assays, percentages of non-DNA-containing particles (alien particles) were determined to avoid an overestimation of sperm-events in the first quadrant (q1) [46]. Briefly, at each relevant time point, 5 µL of each sperm sample were diluted in 895 µL of milliQ^®^-distilled water. Samples were subsequently stained with Propidium Iodide (PI) at a final concentration of 12 µM and incubated at 37.5 °C for 3 min. Percentages of alien particles (f) were used to correct the percentages of non-stained sperm (q_1_) in each sample and staining protocol, according to the following formula:(1)$$q_{1}^{’}=\frac{q_{1}-f}{100-f}\times 100$$
where q_1_’ is the percentage of non-stained sperm after correction.

#### 4.6.1. Sperm Viability (SYBR-14/PI)

Sperm membrane integrity was determined using the LIVE/DEAD^®^ Sperm Viability Kit (Molecular Probes, ThermoFisher Scientific; Waltham, MA, USA). This kit employs a mixture of two dyes: 1) SYBR-14, a membrane permeable fluorochrome that stains the sperm head in green (viable spermatozoa); and 2) propidium iodide (PI), a membrane impermeable fluorochrome that only penetrates through disrupted plasma membranes, staining sperm heads in red (non-viable spermatozoa). Briefly, samples were first incubated with SYBR-14 (final concentration: 100 nM) for 10 min at 37.5 °C in darkness, and then with PI (final concentration: 12 µM) for 5 min at 37.5 °C in the dark. Stained samples were evaluated with the flow cytometer, and only those spermatozoa showing positive SYBR-14 staining and negative PI staining (SYBR-14^+^/PI^−^) were considered as viable, i.e., membrane intact sperm. Results are expressed as mean ± SEM (*n* = 10) (Figure 6).

#### 4.6.2. Acrosomal Exocytosis (PNA-FITC/EthD-1)

Acrosome integrity was evaluated using the lectin from *Arachis hypogaea* (peanut agglutinin, PNA) conjugated with fluorescein isothiocyanate (FITC) and ethidium homodimer (3,8-diamino-5-ethyl-6-phenylphenanthridinium bromide; EthD-1), which was used as a vital stain. In brief, samples were incubated with EthD-1 (final concentration: 2.5 µg/mL) at 37.5 °C for 5 min in the dark. Samples were then centrifuged at 2000 × g and 16 °C for 30 s and resuspended with PBS containing 4 mg/mL bovine serum albumin (BSA). Following this, samples were again centrifuged at the aforementioned conditions and then fixed and permeabilized by adding 100 µL ice-cold methanol (100%) for 30 s. Methanol was removed by centrifugation at 2000 × g and 16 °C for 30 s and pellets were resuspended with 250 µL PBS. Finally, samples were stained with PNA-FITC (final concentration: 2.5 µM) at 25 °C for 15 min in the dark, washed twice with PBS at 2000 × g for 30 s and then resuspended in PBS.

Stained samples were evaluated with the flow cytometer and the following four sperm populations were identified [7]: (1) viable sperm with an intact acrosome (PNA-FITC^+^/EthD-1^−^); (2) viable sperm with an exocytosed acrosome (PNA-FITC^−^/EthD-1^−^); (3) non-viable sperm with an intact acrosome (PNA-FITC^+^/EthD-1^+^); and (4) non-viable sperm with an exocytosed acrosome (PNA-FITC^−^/EthD-1^+^). Fluorescence of EthD-1 was detected through FL3, and PNA-FITC fluorescence through FL1. Results are expressed as the percentage of viable sperm with an exocytosed acrosome (PNA-FITC^-^) in relation to the total viable sperm population (EthD-1^-^) (mean ± SEM; *n* = 10) (Figure 7).

#### 4.6.3. Plasma Membrane Lipid Disorder (M540/YO-PRO-1)

Membrane lipid disorder was determined using Merocyanine 540 (M540; Fluka, 63876) and YO-PRO^®^-1 (Y-3603; Molecular Probes, Invitrogen, ThermoFisher Scientific) [47]. In brief, sperm samples were stained with M540 (final concentration: 2.6 µM) and YO-PRO-1 (final concentration: 25 nM), and incubated at 37.5 °C for 10 min, which was used as a vital stain. YO-PRO-1 fluorescence was collected through an FL1 filter, whereas M540 fluorescence was collected through FL3. The following sperm populations were identified [3]: 1) non-viable sperm with low membrane lipid disorder (M540^−^/YO-PRO-1^+^); 2) non-viable sperm with high membrane lipid disorder (M540^+^/YO-PRO-1^−^); 3) viable sperm with low membrane lipid disorder (M540^-^/YO-PRO-1^−^); and 4) viable sperm with high membrane lipid disorder (M540^+^/YO-PRO-1^−^). Results are expressed as the percentage of viable sperm with a high plasma membrane lipid disorder (M540^+^) in relation to the total viable sperm population (YO-PRO-1^−^) sperm (mean ± SEM; *n* = 10) (Figure 8).

#### 4.6.4. Determination of Intracellular Calcium Levels (Fluo3/PI and Rhod5/YO-PRO-1)

Intracellular calcium levels were evaluated with two separate markers (Fluo3 and Rhod5), which have been reported to have more affinity with the calcium stores of the mid-piece (Fluo3) and of the sperm head (Rhod5), respectively [48]. In the first case, sperm were stained with Fluo3-acetomethoxy ester fluorochrome (Fluo3-AM, F-1241; Molecular Probes, Invitrogen; final concentration: 1 µM) in combination with PI (used as a vital stain; final concentration: 12 µM). After the joint addition of the two fluorochromes, sperm samples were incubated at 37.5 °C for 10 min in darkness [49]. Fluo3 fluorescence was collected through FL1 and PI fluorescence through FL3. According to their fluorescence emission, viable (PI^−^) and non-viable (PI^+^) sperm could show either low (Fluo3^−^) and high (Fluo3^+^) intracellular calcium levels. Results are expressed as the percentages of viable sperm with high calcium levels (Fluo3^+^/PI^−^) in relation to the total viable sperm population (PI^−^), and as geometric fluorescence intensity of Fluo3 in viable sperm with high intracellular calcium levels (Fluo3^+^/PI^−^) (mean ± SEM; *n* = 10; Figure 9).

The other calcium marker (Rhod5-N, 2-(6-Amino-3-imino-3H-xanthen-9-yl)benzoic acid methyl ester; final concentration: 5 µM) was combined with YO-PRO-1 (used as a vital stain; final concentration: 25 nM). After the joint addition of both fluorochromes, sperm samples were incubated at 37.5 °C for 10 min in darkness [6]. YO-PRO-1 fluorescence was collected through FL1 and Rhod5 fluorescence through FL3. Results are expressed as percentages of viable sperm with low intracellular calcium levels (Rhod5^-^/YO-PRO-1^−^) and of viable sperm with high intracellular calcium levels (Rhod5^+^/YO-PRO-1^−^). Results are expressed as the percentages of viable sperm with high calcium levels (Rhod5^+^) in relation to the total viable sperm population (YO-PRO-1^−^), and as geometric fluorescence intensity of Rhod5 in viable sperm with high intracellular calcium levels (Rhod5^+^/YO-PRO-1^−^) (mean ± SEM; *n* = 10; Figure 10).

#### 4.6.5. Determination of Mitochondrial Membrane Potential (JC1)

Mitochondrial membrane potential (MMP) was evaluated following the protocol described by Guthrie and Welch [50]. Samples were incubated with JC1 (5,5’,6,6’-tetrachloro-1,1’,3,3’tetraethylbenzimidazolylcarbocyanine iodide (final concentration: 0.3 µM) at 38 °C for 30 min in the dark. Two different emission filters (FL1 and FL2) were used to differentiate two sperm populations: (i) sperm with high MMP (JC1_agg_) and (ii) sperm with low MMP (JC1_mon_). The percentage of sperm with high MMP corresponded to the orange-stained sperm, which appeared in the upper half of FL1 vs. FL2 dot-plots. FL1 spill-over into the FL2 channel was compensated (51.70%). The results are expressed as percentages of JC1_agg_ sperm and the geometric intensity of FL2 fluorescence in the population of JC1_agg_ sperm (mean ± SEM; *n* = 10) (Figure 11).

### 4.7. Statistical Analyses

Statistical analyses were performed using IBM SPSS 25.0 for Windows (IBM Corp., Armonk, NY; USA). Sperm quality and function parameters (sperm motility, sperm kinematic parameters, sperm viability, acrosomal exocytosis, membrane lipid disorder, intracellular calcium levels and mitochondrial membrane potential) were considered as dependent variables, whereas each experiment and incubation treatments using seminal samples from different boars (*n* = 10 per experiment) were treated as biological replicates. All the variables were first tested for normality (Shapiro-Wilk test) and homoscedasticity (Levene test). Data were then evaluated with a linear mixed model (i.e., with repeated measures) followed by the post-hoc Sidak test for pair-wise comparisons. In this model, the intra-subjects factor was the incubation time, and the inter-subjects factor was the treatment (control, 1 mM 2-GBI, 5 mM 2-GBI and 10 mM 2-GBI). Experiments 1 and 2 were analyzed separately.

In all the statistical analyses, the significant level was set at *p* ≤ 0.05. Results are expressed as means ± standard error of the mean (SEM) (*n* = 10).

## 5. Conclusions

In conclusion, HVCN1 channels are essential for the maintenance of viability, motility and kinematics of pig sperm during in vitro capacitation and progesterone-induced acrosomal exocytosis. While a close relationship between HVCN1 activation and mitochondrial membrane potential was observed, HVCN1 channels were not found to be involved in the regulation of Ca^2+^ influx determined through Fluo3 staining (sperm tail). Despite further research being necessary, HVCN1 activation could also modulate Ca^2+^ entrance to the sperm head and prevent premature acrosomal exocytosis during in vitro capacitation of pig sperm.

## Figures and Tables

**Figure 1 ijms-21-03255-f001:**
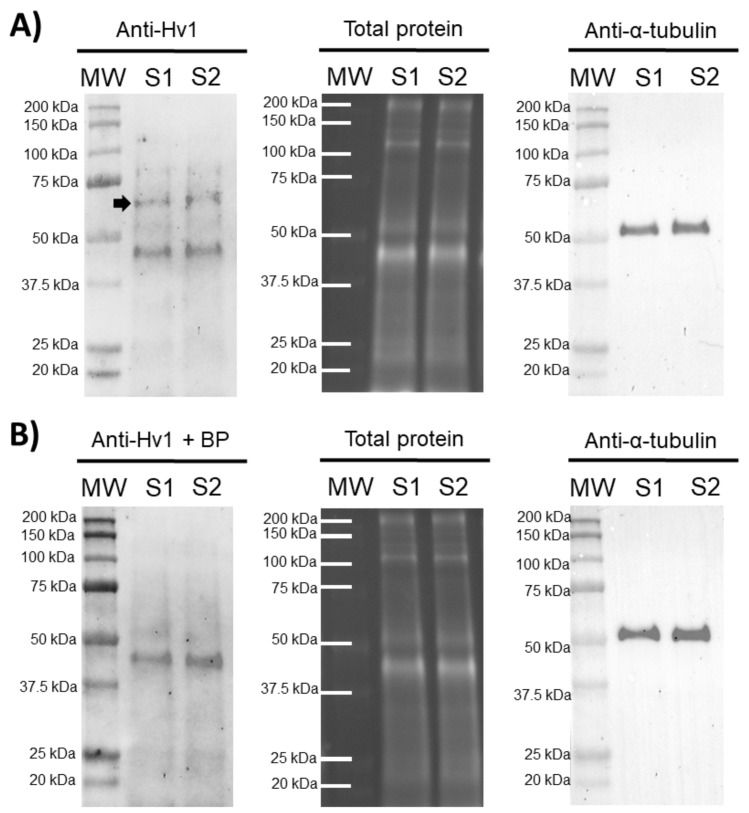
Representative immunoblots for HVCN1 channel. (**A**) Anti-Hv1 in two pig sperm samples (S1 and S2), and its corresponding peptide competition assay; (**B**) Anti-Hv1 + BP. The arrow indicates the HVCN1-specific band of 70 kDa. TGX™ Stain Free (Total protein) and α-tubulin (Anti-α-tubulin) were performed as loading controls.

**Figure 2 ijms-21-03255-f002:**
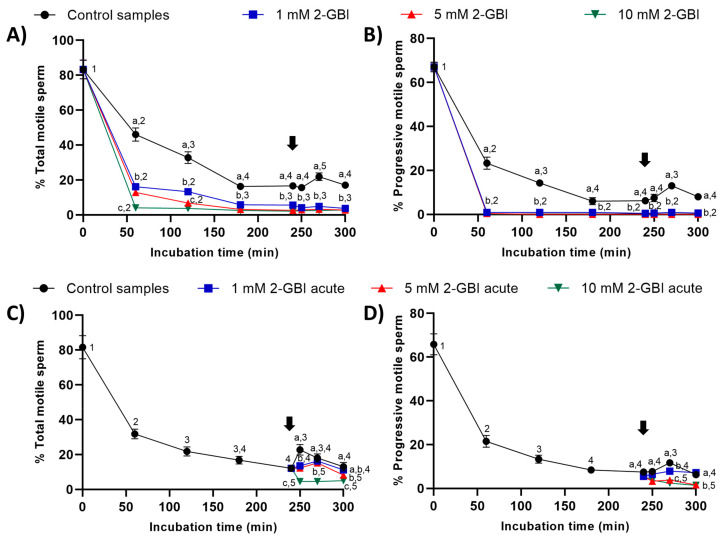
Percentages of total and progressively motile sperm throughout the incubation in capacitating medium (control samples) and capacitating medium plus 1 mM, 5 mM or 10 mM 2-GBI. 2-GBI blocker was added to capacitating medium either at 0 min in experiment 1 (**A**,**B**) or at 240 min in experiment 2 (**C**,**D**; acute). Different superscript letters (a–c) indicate significant differences (*p* < 0.05) between treatments within the same time point, and different superscript numbers (1–5) indicate significant differences (*p* < 0.05) between time points within a given treatment. Results are expressed as mean ± standard error of the mean (SEM) (*n* = 10). The arrow indicates the time at which 10 µg/mL of progesterone was added to all samples (i.e., 240 min of incubation).

**Figure 3 ijms-21-03255-f003:**
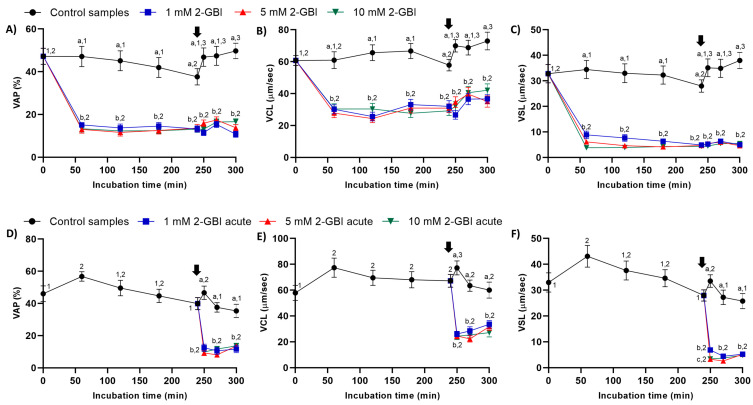
Kinematic parameters of curvilinear velocity (VCL), straight-line velocity (VSL) and average path velocity (VAP) throughout the incubation of sperm in capacitating medium (control samples) and capacitating medium plus 1 mM, 5 mM or 10 mM 2-GBI. 2-GBI blocker was added to capacitating medium either at 0 min in experiment 1 (**A**–**C**) or after 240 min of incubation in experiment 2 (**D**–**F**; acute). Different superscript letters (a–c) indicate significant differences (*p* < 0.05) between treatments within the same time point, and different superscript numbers (1–3) indicate significant differences (*p* < 0.05) between time points within a given treatment. Results are expressed as mean ± SEM (*n* = 10). The arrow indicates the time at which 10 µg/mL of progesterone was added to all samples (i.e., 240 min of incubation).

**Figure 4 ijms-21-03255-f004:**
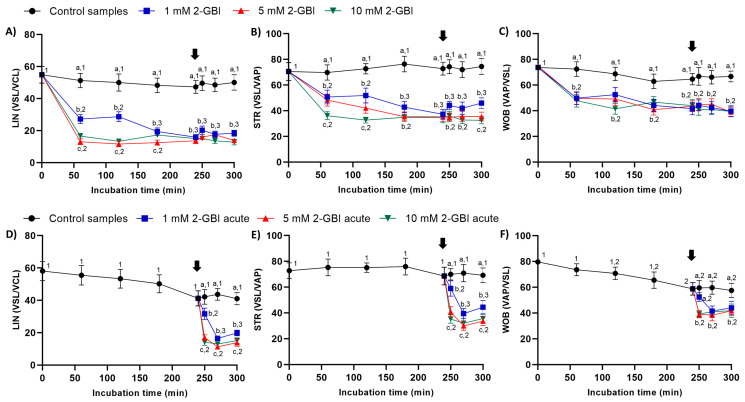
Indexes of linearity (LIN), straightness (STR) and oscillation (WOB) throughout the incubation of sperm in capacitating medium (control samples) and capacitating medium plus 1 mM, 5 mM or 10 mM 2-GBI, added at either 0 min in experiment 1 (**A**–**C**) or 240 min in experiment 2 (**D**–**F**; acute). Different superscript letters (a–c) indicate significant differences (*p* < 0.05) between treatments within the same time point, and different superscript numbers (1–3) indicate significant differences (*p* < 0.05) between time points within a given treatment. Results are expressed as mean ± SEM (*n* = 10). The arrow indicates the time at which 10 µg/mL of progesterone was added to all samples (i.e., 240 min of incubation).

**Figure 5 ijms-21-03255-f005:**
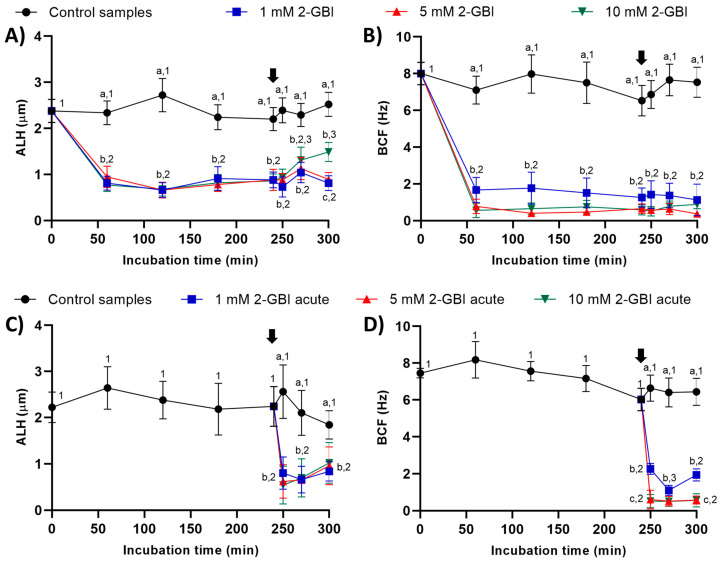
Amplitude of lateral head displacement (ALH) and beat cross frequency (BCF) throughout the incubation of sperm in capacitating medium (control samples) and capacitating medium plus 1 mM, 5 mM or 10 mM 2-GBI. The blocker was added to capacitating medium either at 0 min in experiment 1 (**A**–**C**) or after 240 min of incubation in experiment 2 (**D**–**F**; acute). Different superscript letters (a–c) indicate significant differences (*p* < 0.05) between samples within the same time point, and different superscript numbers (1–3) indicate significant differences (*p* < 0.05) between time points within a given treatment. Results are expressed as mean ± SEM (*n* = 10). The arrow indicates the time at which 10 µg/mL of progesterone was added to all samples (i.e., 240 min of incubation).

**Figure 6 ijms-21-03255-f006:**
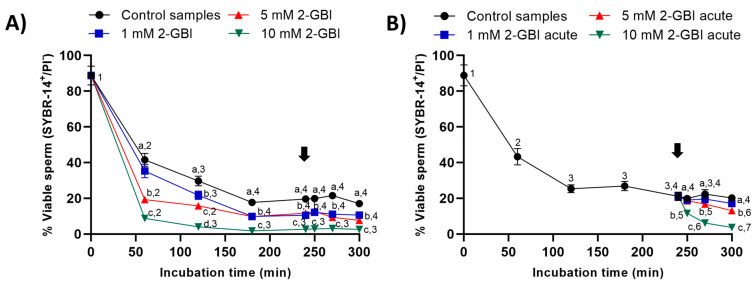
Percentages of viable sperm (SYBR14^+^/PI^-^) throughout the incubation of sperm samples in capacitating medium (control samples) and capacitating medium plus 1 mM, 5 mM or 10 mM 2-GBI. The blocker was added to capacitating medium either at 0 min in experiment 1 (**A**) or after 240 min of incubation in experiment 2 (**B**, acute). Different superscript letters (a–d) indicate significant differences (*p* < 0.05) between samples within the same time point, and different superscript numbers (1–7) indicate significant differences (*p* < 0.05) between time points within a given treatment. Results are expressed as mean ± SEM (*n* = 10). The arrow indicates the time at which 10 µg/mL of progesterone was added to all samples (i.e., 240 min of incubation).

**Figure 7 ijms-21-03255-f007:**
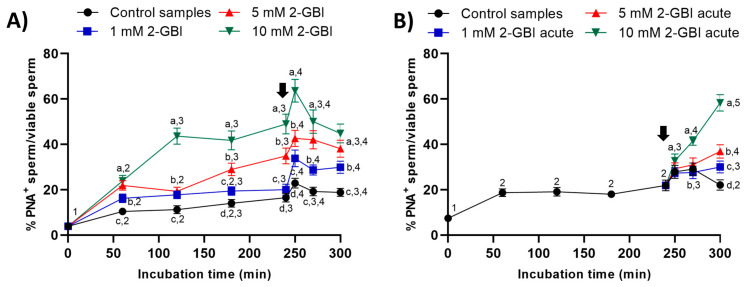
Percentages of viable sperm with exocytosed acrosome (PNA^−^) in relation to total viable sperm (EthD-1^−^) throughout the incubation of sperm samples in capacitating medium (control samples) and capacitating medium plus 1 mM, 5 mM or 10 mM 2-GBI. The blocker was added to capacitating medium either at 0 min in experiment 1 (**A**) or after 240 min of incubation in experiment 2 (**B**, acute). Different superscript letters (a–d) indicate significant differences (*p* < 0.05) between samples within the same time point, and different superscript numbers (1–5) indicate significant differences (*p* < 0.05) between time points within a given treatment. Results are expressed as mean ± SEM (n = 10). The arrow indicates the time at which 10 µg/mL of progesterone was added to all samples (i.e., 240 min of incubation).

**Figure 8 ijms-21-03255-f008:**
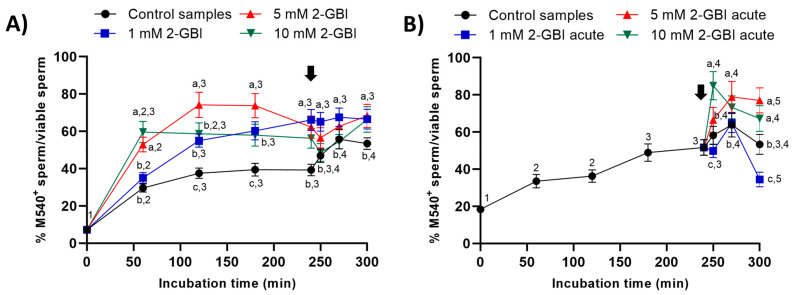
Percentages of viable sperm with high lipid disorder (M540^+^) in relation to total viable sperm (YO-PRO-1^−^) throughout the incubation of sperm samples in capacitating medium (control samples) and capacitating medium plus 1 mM, 5 mM and 10 mM 2-GBI. The inhibitor was added to capacitating medium either at 0 min in experiment 1 (**A**) or after 240 min of incubation in experiment 2 (**B**, acute). Different superscript letters (a–c) indicate significant differences (*p* < 0.05) between samples within the same time point, and different superscript numbers (1–5) indicate significant differences (*p* < 0.05) between time points within a given treatment. Results are expressed as mean ± SEM (*n* = 10). The arrow indicates the time at which 10 µg/mL of progesterone was added to all samples (i.e., 240 min of incubation).

**Figure 9 ijms-21-03255-f009:**
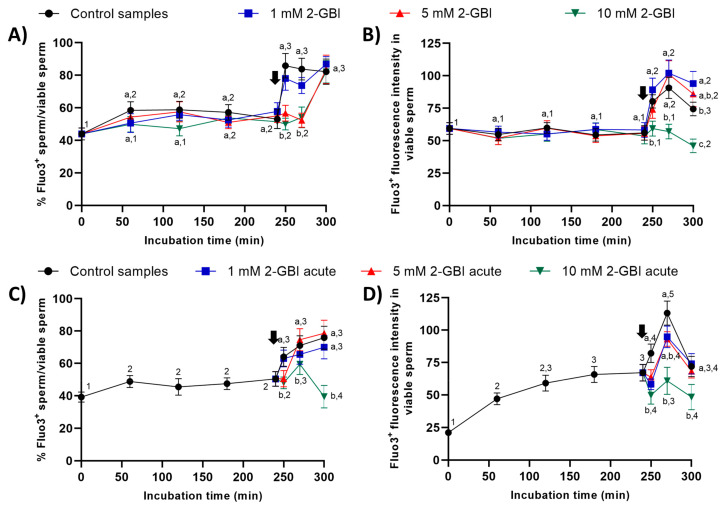
Percentages of viable sperm with high intracellular calcium levels evaluated through Fluo3 staining (Fluo3^+^) in relation to viable sperm (PI^-^), and fluorescence intensity of Fluo3^+^ in viable sperm throughout incubation in capacitating medium (control samples) or capacitation medium plus 1 mM, 5 mM or 10 mM 2-GBI blocker, added at either 0 min in experiment 1 (**A**,**B**) or 240 min in experiment 2 (**C**,**D**; acute). Different superscript letters (a–c) indicate significant differences (*p* < 0.05) between samples within the same time point, and different superscript numbers (1–5) indicate significant differences (*p* < 0.05) between time points within a given treatment. Results are expressed as mean ± SEM (*n* = 10). The arrow indicates the time at which 10 µg/mL of progesterone was added to all samples (i.e., 240 min of incubation).

**Figure 10 ijms-21-03255-f010:**
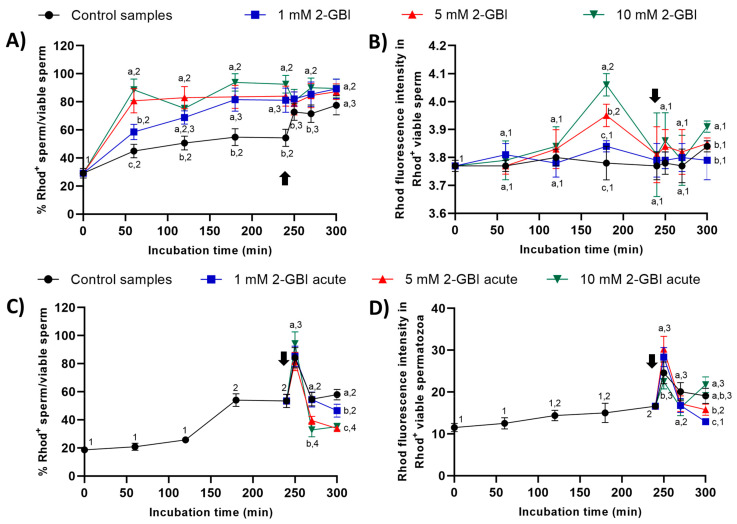
Percentages of viable sperm with high intracellular calcium levels evaluated through Rhod5 (Rhod5^+^) in relation to total viable sperm (YO-PRO-1^−^), and fluorescence intensity of Rhod5^+^ in viable sperm throughout incubation in capacitating medium (control samples) or capacitating medium plus 1 mM, 5 mM or 10 mM, added at either 0 min in experiment 1 (**A**,**B**) or 240 min in experiment 2 (**C**,**D**; acute). Different superscript letters (a–c) indicate significant differences (*p* < 0.05) between samples within the same time point, and different superscript numbers (1–4) indicate significant differences (*p* < 0.05) between time points within a given treatment. Results are expressed as mean ± SEM (*n* = 10). The arrow indicates the time at which 10 µg/mL of progesterone was added to all samples (i.e., 240 min of incubation).

**Figure 11 ijms-21-03255-f011:**
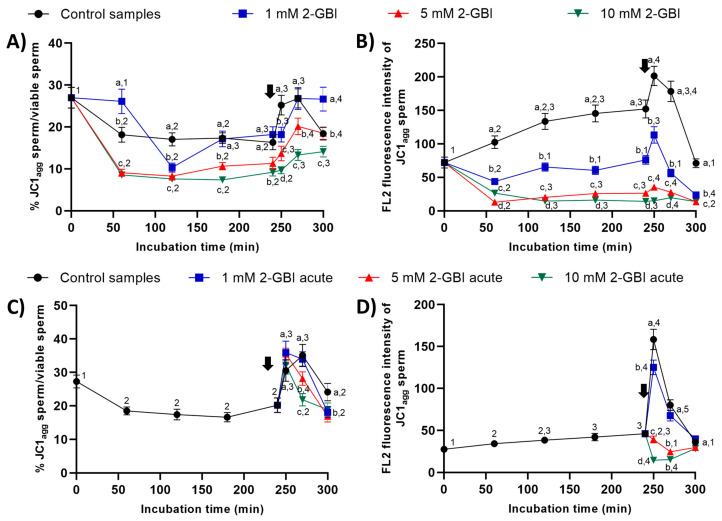
Percentages of sperm with high mitochondrial membrane potential (JC1_agg_), and orange (FL2) fluorescence intensity of JC1_agg_ in sperm with high mitochondrial membrane potential throughout incubation in capacitating medium (control samples) or capacitating medium plus 1 mM, 5 mM or 10 mM 2-GBI. 2-GBI blocker was added either at 0 min in experiment 1 (**A**,**B**) or after 240 min of incubation in experiment 2 (**C**,**D**; acute). Different superscript letters (a–d) indicate significant differences (*p* < 0.05) between samples within the same time point, and different superscript numbers (1–5) indicate significant differences (*p* < 0.05) between time points within a given treatment. Results are expressed as mean ± SEM (*n* = 10). The arrow indicates the time at which 10 µg/mL of progesterone was added to all samples (i.e., 240 min of incubation).

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
