# Peer review of "HVCN1 Channels Are Relevant for the Maintenance of Sperm Motility During In Vitro Capacitation of Pig Spermatozoa"

_ijms, 2020, doi:10.3390/ijms21093255_

Round 1

Reviewer 1 Report

General Comments:

The paper describes the relevance of HVCN1 channels in the maintenance of sperm motility during in vitro capacitation of pig spermatozoa.

The general objective of the manuscript is clear but the secondary objective of each of the proposed experiments is unclear. What do the authors intend to answer each of the proposed experiments?

The "statistical analysis" section is imprecise and would need to be expanded, especially the statistical model used.  Including the information of the results, the applied model would be an experiment (1,2), the time (t0…t5) hierarchized to concentration(1, 5, 10 mM)  and the male as a random effect, is it like this?. Please, clarify. Related to this comment, indicate that the footer of the figures indicate that it "Different superscripts indicate significant differences (p<0.05) between samples within the same time point” but in the presentation of some results it seems that a comparison is made between times point in the same samples (lines 85-89; lines 116-118…). This aspect should be clarified in order to be precise in the result section.

In my opinion, the authors should confirm the interpretation of the results when they compare means between points in the same sample and consider if the discussion should be reviewed

The number of boars used is 20 (line 398), each boar is used in the experimental design (line 410 and 417) but finally n=10 (line 481, 491…). Please, clarify.

It is striking that the control sample of the two experiments seems to show difference in some cases, not only in the values but also in the pattern over time. It is just an appreciation when looking at the figures. For example, motility is greater than 40% in experiment 1 and 50 min but close to 30% in experiment 2 or VAP, VCL and VSL increase from 240 min in experiment 1 but decreases in experiment 2. The control samples for % Fluor3 also seem to be different between both experiments, while in experiment 1 the % is constant from 0 min to 240 min (around 55%), the % is duplicated in experiment 2 from about 25% to 50%.

Specific comments

Line 86: “a slight increase”. There is not significative difference (P<0.05).

Lines 86-89: This information is not in figure 2A. See general comments

Line 111: Not for 10 nM 2-GBI at 270 min.

Lines 116-118. Results did not show in figure

Line 131-132. Results did not show in figure

Line 133-134. With the control for WOB but also between the 2-GBI  concentration for LIN and STR.

Line 145-146. Or maybe a significant decrease of ALH at 300 min for samples incubating with 1 and 5mM 2-GBI. We do not know the information in the figures.

Lines 153-154. See general comment

Line 160-161: Not really. It seems that all samples increased during the first 120 min, after the patron seems to be different among groups. Again, it is not possible to know with the information provided in the figure.

Line 163-166. See general comment

204-206: again the figures show the difference between concentration within a time

Line 209-210. Maybe, there are no differences between 250 min and 270 min for 10mM. I do not know.

Lines 211-212. Not really, at 250 min control group (a) is different from the other samples (b); at 300 min, 10mM concentration is different from the others. The superscripts are not precise at 270 min, it seems that 1 mM and 5 mM are the same points, but one of them is similar to control (a) and the other is similar to 10 mM  (b).

  1. This sentence could be correct at 60 min, but it is imprecise at 120, 180 and 240 min.

Lines 217-218. See general comments

Lines 219-220. Not precise

Line 221-224. Rewrite the paragraph

Author Response

General Comments

The paper describes the relevance of HVCN1 channels in the maintenance of sperm motility during in vitro capacitation of pig spermatozoa.

Answer: We really appreciate the referee for taking their time to review our Manuscript and for their positive comments.

  1. The general objective of the manuscript is clear but the secondary objective of each of the proposed experiments is unclear. What do the authors intend to answer each of the proposed experiments?

Answer: We agree with this comment, so in the first paragraph of the Results section we have clearly indicated the specific objectives of Experiment 1 and Experiment 2.

  1. The "statistical analysis" section is imprecise and would need to be expanded, especially the statistical model used.  Including the information of the results, the applied model would be an experiment (1,2), the time (t0…t5) hierarchized to concentration(1, 5, 10 mM)  and the male as a random effect, is it like this?. Please, clarify. Related to this comment, indicate that the footer of the figures indicate that it "Different superscripts indicate significant differences (p<0.05) between samples within the same time point” but in the presentation of some results it seems that a comparison is made between times point in the same samples (lines 85-89; lines 116-118…). This aspect should be clarified in order to be precise in the result section.

Answer: According to this comment, we have carefully revised the Statistical analysis section and provided detailed information about the which statistical analysis was performed. Herein, we would like to mention that in the model (linear mixed model, i.e. with repeated measures), the boar was not considered as a random-effects factor, because each ejaculate came from a separate boar.

With regard to the original Figures, the reviewer is right since, in order for them not to look overcrowded, we only showed the differences between treatments because they were deemed to be the most relevant ones. However, and following the reviewer’s comment, Figures have been thoroughly revised and both differences between treatments and throughout incubation are now shown. This has been clearly indicated in the Figure legends.

Finally, we have revised carefully the Results section in order to improve its readability and avoid any misunderstanding.

  1. In my opinion, the authors should confirm the interpretation of the results when they compare means between points in the same sample and consider if the discussion should be reviewed

Answer: As indicated in the previous point, all Figures have been revised, and not only differences between treatments but also between incubation time within a given treatment are now shown. We have also revised the Results and Discussion sections in order to improve their quality and readability.

  1. The number of boars used is 20 (line 398), each boar is used in the experimental design (line 410 and 417) but finally n=10 (line 481, 491…). Please, clarify.

Answer: The total number of boars used is 20, 10 in each experiment. In order to avoid misunderstandings, we have clarified this issue in subsection 4.2. Semen samples.

  1. It is striking that the control sample of the two experiments seems to show difference in some cases, not only in the values but also in the pattern over time. It is just an appreciation when looking at the figures. For example, motility is greater than 40% in experiment 1 and 50 min but close to 30% in experiment 2 or VAP, VCL and VSL increase from 240 min in experiment 1 but decreases in experiment 2. The control samples for % Fluor3 also seem to be different between both experiments, while in experiment 1 the % is constant from 0 min to 240 min (around 55%), the % is duplicated in experiment 2 from about 25% to 50%.

Answer: Thank you very much for this comment. As the reviewer knows, biological samples have an inherent variability. Therefore, because the two experiments were performed with semen samples coming from separate boars, they were not totally coincident. This biological variability is assumed to occur, and this is why all results were discussed with regard to the control. We have looked back previous studies conducted in our lab as well as other studies published by other research groups and we have concluded that, while similar trends are observed, data are not identical.

Specific comments

  1. Line 86: “a slight increase”. There is not significative difference (P<0.05).

Answer: In the revised version, we have changed the word “slight” by “non-significant”.

  1. Lines 86-89: This information is not in figure 2A. See general comments

Answer: As previously indicated, in the revised version we have improved the Figures by including superscript numbers, which indicate significant differences between time points within a given treatment. This has been clearly indicated in Figure legends.

  1. Line 111: Not for 10 nM 2-GBI at 270 min.

Answer: We have carefully revised this sentence and we can confirm that it is correct; at 270 min, percentages of progressively motile spermatozoa in Experiment 2 did not differ between 2-GBI concentrations .

  1. Lines 116-118. Results did not show in figure

Answer: Thank you very much for your comment. In response to this and other previous comments, we have improved Figures including superscript numbers that indicate significant differences between time points within a given treatment. We have also corrected Figure legends accordingly.

  1. Line 131-132. Results did not show in figure

Answer: In response to this and other previous comments, we have improved Figures including superscript numbers that indicate significant differences between time points within a given treatment. We have also corrected Figure legends accordingly.

  1. Line 133-134. With the control for WOB but also between the 2-GBI concentration for LIN and STR.

Answer: Sorry, but we do not understand what exactly this reviewer was attempting to say. However, we have carefully revised the Figure and the text and we can confirm they both agree.

  1. Line 145-146. Or maybe a significant decrease of ALH at 300 min for samples incubating with 1 and 5mM 2-GBI. We do not know the information in the figures.

Answer: In accordance with your comment, we have changed this sentence and highlighted that ALH decreased significantly when sperm were incubated with 1 mM and 5 mM 2-GBI. We have also checked that Figures were correct.

  1. Lines 153-154. See general comment

Answer: In response to this and other previous comments, we have improved Figures including superscript numbers that indicate significant differences between time points within a given treatment. We have also corrected Figure legends accordingly.

  1. Line 160-161: Not really. It seems that all samples increased during the first 120 min, after the patron seems to be different among groups. Again, it is not possible to know with the information provided in the figure.

Answer: According to your comment, we have checked our stats and corrected this subsection in order to avoid misunderstandings. Please also note that in the revised version we have improved Figures by including superscript numbers that indicate significant differences between time points within a given treatment. We have also corrected Figure legends accordingly.

  1. Line 163-166. See general comment

Answer: Thank you very much for this comment. Please also note that in the revised version we have improved Figures by including superscript numbers that indicate significant differences between time points within a given treatment. We have also corrected Figure legends accordingly.

  1. Line 204-206: again the figures show the difference between concentration within a time

Answer: In response to this and other previous comments, we have improved Figures including superscript numbers that indicate significant differences between time points within a given treatment. We have also corrected Figure legends accordingly.

  1. Line 209-210. Maybe, there are no differences between 250 min and 270 min for 10mM. I do not know.

Answer: We checked statistical results and we confirm that samples treated with 10 mM GBI and evaluated at 250 min significantly differed from those evaluated at 270 min.

  1. Lines 211-212. Not really, at 250 min control group (a) is different from the other samples (b); at 300 min, 10mM concentration is different from the others. The superscripts are not precise at 270 min, it seems that 1 mM and 5 mM are the same points, but one of them is similar to control (a) and the other is similar to 10 mM (b).

Answer: We completely agree with your comment. Therefore, we have carefully revised the content of these lines to explain clearly the changes in fluorescence intensity in the different treatments.

  1. Line 216. This sentence could be correct at 60 min, but it is imprecise at 120, 180 and 240 min.

Answer: We checked the sentence of line 216, but we think that there was a typographic mistake and the reviewer rather referred to line 215. We complete agree with the reviewer’s comment; for this reason, we have revised and improved the description of the results obtained from the Rhod assay in order to provide a clear explanation of which changes occurred in this parameter from 120 to 240 min of incubation.

  1. Lines 217-218. See general comments

Answer: Please check that, in the revised version, we have improved the Figures by indicating the significant differences between time points within a given treatment. We have also corrected Figure legends accordingly.

  1. Lines 219-220. Not precise

Answer: We completely agree with your comment. Therefore, we have carefully revised this sentence in order to give a detailed information about changes occurring in the geometric mean intensity of Rhod5 fluorescence throughout Experiment 1.

  1. Line 221-224. Rewrite the paragraph

Answer: We completely agree with your comment. Therefore, we have carefully revised and rewritten this paragraph in order to provide a detailed information on changes in Rhod5 fluorescence intensity (Experiment 2).

Reviewer 2 Report

This manuscript studies the physiological role of HVCN1 channels during boar sperm in vitro capacitation. Sperm motility and kinematics, acrosome exocytosis, membrane lipid disorder, intracellular calcium levels and mitochondrial membrane potential were studied.

I find this research paper very complete and interesting. The experimental design and material and methods are correct, and the conclusions are in accordance with the results. However, several minor issues must be solved before the manuscript publication.

  • Please, define HVCN1 in the abstract
  • Figures should be near the first time they are cited. Please, ensure all that the figures are placed after the paragraph they are mentioned.
  • Significant differences in progressive motility after progesterone addition are not indicated in the figures. Maybe you could use a symbol (like *) to show significant differences before and after progesterone treatment in the experimental samples.
  • If there are no significant differences, and it is indicated in the text, it is not necessary to add p > 0.05. Please, delete it in lines 112, 117 and others.
  • In lines 131-132 you said that “The addition of progesterone after 240 min of incubation had little effect on those indexes in both control and blocked samples”, but then you indicated that p < 0.05. Please, clarify this.
  • Lines 183-184 indicate that “In experiment 2, the addition of progesterone together with that of 1 mM 2-GBI induced a significant increase (p > 0.05) in the percentage of viable sperm…” I think there is a mistake here. Please, clarify it.
  • In the material and methods section, you indicate that 0 hours is t0 and 240 minutes t4, and more, but you did not use this notation in the results section or the figures. Please, delete them.
  • Please, indicate the n of your study in the figures.
  • Please, explain how did you block the membranes in the immunoblotting
  • The flow cytometry results are expressed in relation to the total viable sperm population, so it is necessary to demonstrate that the addition of the HVCN1 blocker did not affect sperm viability. Please, include a figure showing the viability of the samples during experiment 1 and 2. It can be added as a supplementary figure.

In conclusion, I found this paper suitable for future publication, but all these minor questions must be correctly addressed.

Author Response

This manuscript studies the physiological role of HVCN1 channels during boar sperm in vitro capacitation. Sperm motility and kinematics, acrosome exocytosis, membrane lipid disorder, intracellular calcium levels and mitochondrial membrane potential were studied.

I find this research paper very complete and interesting. The experimental design and material and methods are correct, and the conclusions are in accordance with the results. However, several minor issues must be solved before the manuscript publication.

  • Please, define HVCN1 in the abstract

Answer: We completely agree with this comment. Therefore, we have added the definition of HVCN1 in the Abstract section.

  • Figures should be near the first time they are cited. Please, ensure all that the figures are placed after the paragraph they are mentioned.

Answer: We agree with you that the close proximity between Figures and its corresponding paragraph improves the understanding of the Results. We have done much effort to place each Figure as closer as possible to its corresponding paragraph.

  • Significant differences in progressive motility after progesterone addition are not indicated in the figures. Maybe you could use a symbol (like *) to show significant differences before and after progesterone treatment in the experimental samples.

Answer: Please, check that in the revised version we have improved the Figures by highlighting the significant differences not only between treatments within a given time point, but also between time points within a given treatment. We have also corrected Figure legends accordingly.

  • If there are no significant differences, and it is indicated in the text, it is not necessary to add p > 0.05. Please, delete it in lines 112, 117 and others.

Answer: According to this comment, we have deleted p > 0.05 from the Results section.

  • In lines 131-132 you said that “The addition of progesterone after 240 min of incubation had little effect on those indexes in both control and blocked samples”, but then you indicated that p < 0.05. Please, clarify this.

Answer: We have revised this sentence and checked that the effects of progesterone addition were not significant for any of these kinematics parameters. We have corrected the sentence in order to avoid misunderstandings.

  • Lines 183-184 indicate that “In experiment 2, the addition of progesterone together with that of 1 mM 2-GBI induced a significant increase (p > 0.05) in the percentage of viable sperm…” I think there is a mistake here. Please, clarify it.

Answer: We have revised the sentence and checked that progesterone addition induced a significant increase in the percentage of viable sperm with high membrane lipid disorder at 270 min. Therefore, we have changed “p > 0.05” to “p < 0.05”.

  • In the material and methods section, you indicate that 0 hours is t0 and 240 minutes t4, and more, but you did not use this notation in the results section or the figures. Please, delete them.

Answer: In agreement with this criticism, we have deleted the notation from t0 and t5. We agree with the reviewer that it could be confusing.

  • Please, indicate the n of your study in the figures.

Answer: We have indicated the “n” in Figure legends.

  • Please, explain how did you block the membranes in the immunoblotting

Answer: We have revised the subsection 4.4. Immunoblotting and a more detailed explanation about membrane blocking has been given.

  • The flow cytometry results are expressed in relation to the total viable sperm population, so it is necessary to demonstrate that the addition of the HVCN1 blocker did not affect sperm viability. Please, include a figure showing the viability of the samples during experiment 1 and 2. It can be added as a supplementary figure.

Answer: We also had sperm viability data, but because we already showed results from PNA/EthD-1 staining, we thought that they did not need to be included. However, following the reviewer’s comment, we have revised the Manuscript and added SYBR14/PI data (2.4. Sperm viability) in a new Figure (Figure 6). Please note that we have also added a new subsection in the Material and Methods section (4.6.1. Sperm viability), which contains a detailed description on the protocol used to analyse this sperm parameter.

Please note that in the present study we observed a decrease in the viability, as well as in other sperm parameters, of spermatozoa incubated in capacitation medium. These results obtained in control samples (i.e. incubated in capacitation medium without inhibitor) agree with results previously reported in other studies (Publicover et al., 2007. Nat Cell Biol; Ded et al., 2010. Reprod Biol Endocrinol; Puigmulé et al. 2011. Reprod Fertil Dev; Smith et al., 2013. Proc. Natl. Acad. Sci. USA; Yeste et al., 2015. Andrology; Zou et al., 2017. Hum Reprod; Yeste et al., 2019. Int J Mol Sci). Therefore, the decrease of sperm viability is an inherent effect associated to both in vivo and in vitro capacitation. The novelty of the present study is that the inhibition of HVCN1 channels manifested in a significant decrease in sperm viability, due to the inability of the sperm cells to regulate the internal pH during capacitation. Nevertheless, according to your suggestion, we have carefully revised the Discussion section and highlighted that the results obtained in control samples are in agreement with previous studies. According to the results obtained from blocking experiments, we have also discussed the physiological relevance of HCVN1 channels in the maintenance of sperm viability during in vitro capacitation of pig sperm.

On the other hand, we would also highlight the relevance of expressing the results of flow cytometry in relation to the total viable sperm population, since only viable spermatozoa are able to fertilize the oocyte. Therefore, the analysis of the physiological changes occurring in the total viable sperm population leads us to analyse the real change of viable spermatozoa to undergo the sequence of events associated to in vitro capacitation and their fertilising ability. In the Discussion section of the revised manuscript, we have clarified this point in order to avoid misunderstandings.

In conclusion, I found this paper suitable for future publication, but all these minor questions must be correctly addressed.

Answer: We want to thank you for your comments, because we sincerely think that they have been contributed to improve the quality of the Manuscript.

Reviewer 3 Report

Very interesting paper, well written, well done. 

Author Response

Very interesting paper, well written, well done. 

Answer: We sincerely appreciate your time and favourable evaluation.

Reviewer 4 Report

The study entitled “HVCN1 channels are relevant for the maintenance of sperm motility during in vitro capacitation of pig spermatozoa” investigated the role of the HVCN1 voltage-gated cation channels on sperm capacitation (namely, on sperm total and progressive motility, kinematics and mitochondrial membrane potential). The topic assessed by the Authors is not completely original since a previous study a previous study showed the possible involvement of these channels in the acquisition of human sperm hypermotility (J Cell Sci. 2020 Jan 30;133(2). pii: jcs238816. doi: 10.1242/jcs.238816). However, few data are available on this matter and this is the first study clearly showing the effects of the channel inhibition on sperm function.

Please consider the following comments.

  1. How many times was each experiment performed?
  2. Figure 1. Why is the 45 kDa band visible? What does it indicate?
  3. Line 73. How did the Authors choice the doses of the 2-GBI blocker?
  4. Figure 2. Why did the Authors decide to show the progressive/total motile sperm percentage? What is its importance on a physio-pathological point of view? The Authors should show the progressive motile sperm
  5. Lines 85-89. How do the Authors interpret the ability of the blocker to hinder progesterone-induced increase in total motility and the blocker-induced increase in progressive motility?

Author Response

The study entitled “HVCN1 channels are relevant for the maintenance of sperm motility during in vitro capacitation of pig spermatozoa” investigated the role of the HVCN1 voltage-gated cation channels on sperm capacitation (namely, on sperm total and progressive motility, kinematics and mitochondrial membrane potential). The topic assessed by the Authors is not completely original since a previous study a previous study showed the possible involvement of these channels in the acquisition of human sperm hypermotility (J Cell Sci. 2020 Jan 30;133(2). pii: jcs238816. doi: 10.1242/jcs.238816). However, few data are available on this matter and this is the first study clearly showing the effects of the channel inhibition on sperm function.

Answer: We would like to thank your comments because they have contributed to the improvement of the quality of our Manuscript. We also looked at the cited reference, which has been included in the revised Manuscript since we found it very relevant to our work.

Please consider the following comments.

  1. How many times was each experiment performed?

Answer: Each experiment was repeated 10 times, and each used a different semen sample coming from a separate boar. In the revised version of the manuscript we have clearly indicated that each experiment included 10 seminal samples, each coming from a different boar. Please, check subsection 4.2. Semen samples.

  1. Figure 1. Why is the 45 kDa band visible? What does it indicate?

Answer: The 45 kDa band corresponds to unspecific bond of the antibody. Please note that, in the Results section, this band was shown to disappear after peptide competition assay (see subsection 2.1. Identification of HVCN1 channels).

  1. Line 73. How did the Authors choice the doses of the 2-GBI blocker?

Answer: In subsection 4.3. Experimental design, we clearly indicated that the blocker concentration was set following preliminary experiments and after reviewing the literature.

  1. Figure 2. Why did the Authors decide to show the progressive/total motile sperm percentage? What is its importance on a physio-pathological point of view? The Authors should show the progressive motile sperm

Answer: In agreement to this comment, we provide the results of progressive sperm motility. Therefore, we have revised the content of subsection 2.3.2. Total and progressive motility and provided new graphs for Figure 2.

  1. Lines 85-89. How do the Authors interpret the ability of the blocker to hinder progesterone-induced increase in total motility and the blocker-induced increase in progressive motility?

Answer: According to your comment, we have carefully revised the content of this paragraph and checked that there were no inconsistencies. Please also note that, in agreement with your previous comment, we have also corrected this subsection by providing data as percentages of progressively motile spermatozoa rather than as progressive/total motility ratios. This change has undoubtedly improved the quality and readability of this subsection.

Reviewer 5 Report

This is a very interesting and important update on proton regulation and homeostasis during pig sperm capacitation. The authors in their study aim to decipher the role of HVCN1 channel in pig spermatozoa and how this channel is engaged in the events associated to capacitation. The authors report direct involvement of HVCN1 in sperm motility, and kinematics; however, to this reviewer’s surprise HVCN1 does not seem to be directly involved in Ca2+ regulation via channels in sperm tails like it is in human spermatozoa, signifying the importance of intraspecies comparative studies. Furthermore, the authors report that pH imbalance leads to an acceleration of capacitation associated evens resulting in degenerative acrosomal exocytosis.  This reviewer is impressed by the quality, the soundness and the extend of this study, and suggests minor edits only for the reader to better comprehend the study and to navigate themselves through it easier.

Abstract

  1. L28: it had little effect on the sperm tail Ca2+ levels
  2. L30: and the sperm head Ca2+ levels

Introduction

  1. Please, add a paragraph about sperm capacitation including the events associated to sperm capacitation for the reader to understand the rationale of the study. This will also make the study more comprehensive for more general readers without the background in sperm biology. The first paragraph of the discussion section would be a better fit for the introduction, as this is the first time authors explain why they studied those particular evets associated to capacitation.

Results

  1. L71: acrosomal exocytosis, please correct throughout the whole MS.
  2. L75: have the authors tried to immunolocalize HVCN1 channels in pig spermatozoa? this would improve the MS and might give an additional piece of explanation as to why the authors are observing such results during the inhibition studies.
  3. L219-220, fluorescence intensity seems to be significantly different between the control, 5 mM and 10 mM blocker at 180 min, according to the fig 9A, right

Discussion

  1. L267-70: as per #3 please move this to the introduction
  2. L338-40: In this reviewer’s opinion, Ca2+ entrance to the flagellum does not rely solely on the activity of HVCN1, and the proton extrusion might be compensated for by other channels (e.g. Na+/H+). This reviewer would also expect progesterone to be involved in the calcium uptake in capacitated spermatozoa through a non-direct mechanism rather than directly stimulating calcium channels. These are just the reviewer’s thoughts and curiosity without a need for implementation, as every author is fully entitled to postulate their hypotheses.
  3. L347: the authors mention the possibility of other proton channels engagement, very good.
  4. L349: this is why it would be of interest to look into the HVCN1 localization.
  5. L366-9: The authors in [28] used Zinc to block HVCN1, and Zinc was also shown to inhibit the ubiquitin-proteasome system, acrosin, and MMP2 activities (see https://www.mdpi.com/1422-0067/21/6/2121), which may also be contributors to acrosomal exocytosis. At least inhibition of proteasome was found to inhibit progesterone-induced AE and sustained phase of the Ca2+ influx (Morales, et al., 2003, Hum Reprod 18:1010-7).

Materials and Methods

  1. L447: incubated at 95°C, boiling suggests it was at 100°C, provided it was water
  2. L459: please indicate what stripping reagent was used
  3. L503: Side scatter is usually abbreviated as SSC
  4. L520: Please, state that EHD is a vital dye used instead of PI that can withstand acrosomal staining procedure, unlike PI, so that the reader does not have to look up reference [41] that references Cooper and Yeung, 1998. Out of curiosity, is this how you can differentiate between induced and spontaneous acrosomal loss?
  5. L573: Same as my previous comment, please state that Yo-Pro-1 was chosen as a viable stain since PI has the same excitation/emission characteristics as merocyanine 540 to make it easier for the reader to follow.
  6. Please state that all the ethical guidelines for animal welfare and handling were followed.

Figures

  1. In general, please designate each part (immunoblot, graph) with a letter and add the reference to the text, this will greatly improve the navigation of the reader throughout the study
  2. Figure 1: Please designate Anti-HVCN1 immunoblot with A and peptide blocking study with B and reference in the text.
  3. Figure 2: Please designate the graphs showing the % progressive/total motile sperm for experiment 1 with A’, and experiment 2 with B’. Please reference both in the text.
  4. Figure 3: Please designate the VCL and VSL graphs in experiment 1 as A’ and A”, respectively; and in experiment 2 as B’ and B”, respectively. Please reference all in the text.
  5. Figure 4: Please designate the STR and WOB graphs in experiment 1 as A’ and A”, respectively; and in experiment 2 as B’ and B”, respectively. Please reference all in the text.
  6. Figure 5: Please designate the BCF graphs in experiment 1 as A’, and experiment 2 as B’, respectively. Please reference both in the text.
  7. Figure 8: Please designate the Fluo3+ fluorescence intensity in viable sperm graphs as A’ in experiment 1, and B’ in experiment 2, respectively, and reference both in the text.
  8. Figure 9: Please designate the Rhod5+ fluorescence intensity in viable sperm graphs as A’ in experiment 1, and B’ in experiment 2, respectively, and reference both in the text.
  9. Figure 10: Please designate the FL2 fluorescence intensity of JC1agg sperm graphs as A’ in experiment 1, and B’ in experiment 2, respectively, and reference both in the text.

Author Response

This is a very interesting and important update on proton regulation and homeostasis during pig sperm capacitation. The authors in their study aim to decipher the role of HVCN1 channel in pig spermatozoa and how this channel is engaged in the events associated to capacitation. The authors report direct involvement of HVCN1 in sperm motility, and kinematics; however, to this reviewer’s surprise HVCN1 does not seem to be directly involved in Ca2+ regulation via channels in sperm tails like it is in human spermatozoa, signifying the importance of intraspecies comparative studies. Furthermore, the authors report that pH imbalance leads to an acceleration of capacitation associated evens resulting in degenerative acrosomal exocytosis.  This reviewer is impressed by the quality, the soundness and the extend of this study, and suggests minor edits only for the reader to better comprehend the study and to navigate themselves through it easier.

Answer: We really appreciate the referee for taking their time to review our Manuscript and for their positive comments.

Abstract

  1. L28: it had little effect on the sperm tail Ca2+ levels

Answer: In agreement with your comment, “little” has been changed to “non significant” in the revised version.

  1. L30: and the sperm head Ca2+ levels

Answer: Following you comment, we have revised and corrected this sentence in the revised version.

Introduction

  1. Please, add a paragraph about sperm capacitation including the events associated to sperm capacitation for the reader to understand the rationale of the study. This will also make the study more comprehensive for more general readers without the background in sperm biology. The first paragraph of the discussion section would be a better fit for the introduction, as this is the first time authors explain why they studied those particular evets associated to capacitation.

Answer: In agreement with your comment, we have moved the first paragraph of the Discussion to the Introduction section.

Results

  1. L71: acrosomal exocytosis, please correct throughout the whole MS.

Answer: In accordance with your suggestion, we have change “acrosome exocytosis” by “acrosomal exocytosis” in the revised version of the manuscript

  1. L75: have the authors tried to immunolocalize HVCN1 channels in pig spermatozoa? this would improve the MS and might give an additional piece of explanation as to why the authors are observing such results during the inhibition studies.

Answer: We tried to immunolocalize the HVCN1 channels in boar sperm, but we had technical problems due to antibody specificity. Because of COVID-19 pandemic, we cannot solve these difficulties right now, and we are afraid that we won’t be able to do so for a long time. However, since the present study is focused on determining the physiological role of HVCN1 channels during in vitro capacitation, we considered that immunolocalization studies were not essential.

  1. L219-220, fluorescence intensity seems to be significantly different between the control, 5 mM and 10 mM blocker at 180 min, according to the fig 9A, right

Answer: In accordance with your comment, we have carefully revised these lines and improved the description of the results obtained. Please check subsection 2.7.2. Rhod5 staining.

Discussion

  1. L267-70: as per #3 please move this to the introduction

Answer: In accordance with this and with your previous comments, we have moved these lines to the Introduction section of the revised manuscript.

  1. L338-40: In this reviewer’s opinion, Ca2+ entrance to the flagellum does not rely solely on the activity of HVCN1, and the proton extrusion might be compensated for by other channels (e.g. Na+/H+). This reviewer would also expect progesterone to be involved in the calcium uptake in capacitated spermatozoa through a non-direct mechanism rather than directly stimulating calcium channels. These are just the reviewer’s thoughts and curiosity without a need for implementation, as every author is fully entitled to postulate their hypotheses.

Answer: We have carefully revised the content of these lines and, in agreement with your suggestion, we have changed our hypothesis by postulating that Ca2+ entrance to the flagellum may be dependent of both direct and non-direct effect of progesterone on Ca2+ channels.

  1. L347: the authors mention the possibility of other proton channels engagement, very good.

Answer: Thank you very much for your comment.

  1. L349: this is why it would be of interest to look into the HVCN1 localization.

Answer: We completely agree with your suggestion. However, as we already indicated, it is impossible for us to perform such experiments. Moreover, we consider that, although HVCN1 localisation is of great interest, it is not essential for the present study. Additionally, and agreeing with you regarding the importance of HVCN1 localisation, we are designing a new study in which we will inquire in this regard.

  1. L366-9: The authors in [28] used Zinc to block HVCN1, and Zinc was also shown to inhibit the ubiquitin-proteasome system, acrosin, and MMP2 activities (see https://www.mdpi.com/1422-0067/21/6/2121), which may also be contributors to acrosomal exocytosis. At least inhibition of proteasome was found to inhibit progesterone-induced AE and sustained phase of the Ca2+ influx (Morales, et al., 2003, Hum Reprod 18:1010-7).

Answer: We have read the papers that you cited in your comment and used them to improve the Discussion in these lines.

Materials and Methods

  1. L447: incubated at 95°C, boiling suggests it was at 100°C, provided it was water

Answer: Following your suggestion, the word “boiled” was changed to “incubated”.

  1. L459: please indicate what stripping reagent was used

Answer: As you kindly requested, stripping reagents and protocol details were added within the text.

  1. L503: Side scatter is usually abbreviated as SSC

Answer: In accordance with your comment, we have change the abbreviation “SS” to “SSC” in the revised manuscript

  1. L520: Please, state that EHD is a vital dye used instead of PI that can withstand acrosomal staining procedure, unlike PI, so that the reader does not have to look up reference [41] that references Cooper and Yeung, 1998. Out of curiosity, is this how you can differentiate between induced and spontaneous acrosomal loss?

Answer: Following your comment, we have indicated that EthD-1 was used as a vital dye. The reason why this test was performed is that PNA/PI without permeabilization does not allow to really determine which sperm population are viable and exhibit an exocytosed acrosome (induced acrosome exocytosis). Although one cannot be completely sure, this would allow differing from sperm with spontaneous acrosomal loss (PNA-). In addition, the reason why EthD-1 was used as a vital dye instead of PI is related to the recommendations of Cooper and Yeung (1998), who indicated that, for flow cytometry experiments and after sperm permeabilization, fluorescence from EthD-1 was more stable than that of PI.

  1. L573: Same as my previous comment, please state that Yo-Pro-1 was chosen as a viable stain since PI has the same excitation/emission characteristics as merocyanine 540 to make it easier for the reader to follow.

Answer: In accordance with your comment, in the revised version we have added that YO-PRO-1 was used as a vital dye.

  1. Please state that all the ethical guidelines for animal welfare and handling were followed.

Answer: In agreement with your comment, we have stated in the revised manuscript that all the ethical guidelines for animal welfare and handling were followed; please check subsection 4.2. Semen samples. However, and we would also like to mention that because seminal samples were purchased from a local farm and we did not manipulate any animal, a specific authorization from an Ethics Committee was not required.

Figures

  1. In general, please designate each part (immunoblot, graph) with a letter and add the reference to the text, this will greatly improve the navigation of the reader throughout the study

Answer: Following your suggestion, we have designated all the graphs with a letter.

  1. Figure 1: Please designate Anti-HVCN1 immunoblot with A and peptide blocking study with B and reference in the text.

Answer: In Figure 1, AntiHVCN1 and peptide blocking study have been respectively designated as (A) and (B), as you kindly requested.

  1. Figure 2: Please designate the graphs showing the % progressive/total motile sperm for experiment 1 with A’, and experiment 2 with B’. Please reference both in the text.

Answer: According to your suggestion we have designated all the graphs using A, B, C and D.

  1. Figure 3: Please designate the VCL and VSL graphs in experiment 1 as A’ and A”, respectively; and in experiment 2 as B’ and B”, respectively. Please reference all in the text.

Answer: According to your suggestion we have designated all the graphs using A, B, C and D.

  1. Figure 4: Please designate the STR and WOB graphs in experiment 1 as A’ and A”, respectively; and in experiment 2 as B’ and B”, respectively. Please reference all in the text.

Answer: According to your suggestion we have designated all the graphs using A, B, C and D.

  1. Figure 5: Please designate the BCF graphs in experiment 1 as A’, and experiment 2 as B’, respectively. Please reference both in the text.

Answer: According to your suggestion we have designated all the graphs using A, B, C and D.

  1. Figure 8: Please designate the Fluo3+ fluorescence intensity in viable sperm graphs as A’ in experiment 1, and B’ in experiment 2, respectively, and reference both in the text.

Answer: According to your suggestion we have designated all the graphs using A, B, C and D.

  1. Figure 9: Please designate the Rhod5+ fluorescence intensity in viable sperm graphs as A’ in experiment 1, and B’ in experiment 2, respectively, and reference both in the text.

Answer: According to your suggestion we have designated all the graphs using A, B, C and D.

  1. Figure 10: Please designate the FL2 fluorescence intensity of JC1agg sperm graphs as A’ in experiment 1, and B’ in experiment 2, respectively, and reference both in the text.

Answer: According to your suggestion we have designated all the graphs using A, B, C and D.

Round 2

Reviewer 1 Report

Dear authors,

You have made really good improvements in text and figures, and the quality has really improved, especially in results sections.